



# On the influence of cross-sectional deformations on the aerodynamic performance of wind turbine rotor blades

Julia Gebauer[1], Felix Prigge[1], Dominik Ahrens[2], Lars Wein[2], and Claudio Balzani[1]

[1]Institute for Wind Energy Systems, Leibniz University Hannover, Appelstr. 9A, 30167 Hanover, Germany
[2]Institute of Turbomachinery and Fluid Dynamics, Leibniz University Hannover, An der Universität 1, 30823 Garbsen, Germany

**Correspondence:** Julia Gebauer (research@iwes.uni-hannover.de)

**Abstract.** The aerodynamic performance of a wind turbine rotor blade depends on the geometry of the used airfoils. The airfoil shape can be affected by elastic deformations of the blade during operation due to structural loads. This paper provides an initial estimation of the extent to which cross-sectional deformations influence the aerodynamic load distribution along the rotor blade. The IEA 15 MW reference wind turbine model is used for this study. A constant wind field at the rated wind speed is applied as a test case. The resulting loads are calculated by an aero-servo-elastic simulation of the turbine. The loads are applied to a 3D finite element (FE) model of the rotor blade, which serves to calculate the cross-sectional deformations. For the individual cross-sections in the deformed configuration, the new lift and drag coefficients are calculated. These are then included in the aero-servo-elastic simulation and the obtained results are compared with those of the initial simulation that is based on the undeformed cross-sections. The cross-sectional deformations consist of a change in the chord length and the geometry of the trailing edge panels and depend largely on the azimuth position of the blade. The change in the airfoil geometries results in altered aerodynamic characteristics and therefore in a deviation of the blade root bending moments, the maximum change of which is -1.4% in the in-plane direction and +0.71% in the out-of-plane direction. Although these values are relatively small, the initial results imply that further investigations should be carried out with more complex wind fields and different rotor blade designs to identify aero-structural couplings that may be critical for the design of rotor blades or other wind turbine components.

## 1 Introduction

The rotor blade is a crucial element in the generation of electrical power from wind in modern wind turbines. The blades are exposed to a wide range of loads during their life time, which are a combination of aerodynamic, gravitational, inertial, transient, and gyroscopic loads (Hau, 2013; Söker, 2013; Liu et al., 2017; Burton et al., 2021). Especially the wind inflow and the blade mass contribute significantly to out-of-plane (OOP) and in-plane (IP) bending moments.

The amount of energy that can be extracted from the wind depends on the aerodynamic design of the rotor blade, i.e., the shape of the outer shell. The aerodynamic design is a sequence of airfoils that are threaded along



the blade axis. A high lift-to-drag ratio in each individual airfoil is desirable for maximum power generation. The geometry of the airfoil and the angle of attack, which is calibrated by the aerodynamic twist angle, are crucial for a high aerodynamic performance. However, a high power output is normally accompanied by high aerodynamic and consequentially high mechanical loads acting on the blade. Thus, a compromise must always be found in the blade design between aerodynamic and structural performance (Hansen, 2015; Bak, 2023).

The structural design of a rotor blade provides high stiffness and strength with respect to its weight. In the context of this paper, the structural composition of the blade defines its resistance against cross-sectional deformations, and thus the resistance against changes of the blade geometry and the resulting aerodynamic performance. The cross-sectional stiffness depends on the choice of materials, the structural topology, and the layup of the composite structures (Schürmann et al., 2007; Vassilopoulos, 2013). With currently used materials and classical blade topologies

and layups, and the tendency towards larger rotor blades, the weight, the generated power, and thus the structural loads increase. When optimising the design with the minimisation of the blade mass as an optimisation target, an increasingly elastic behaviour of the blade is expected, including the elasticity and flexibility of the cross-sections.

Several authors have analysed the cross-sectional deformation of thin-walled structures, which can be divided into in-plane and out-of-plane deformations (warping). Among these works, Gabriele et al. (2016) presented a model

based on shell theory that is able to describe the in-plane cross-sectional deformations. The analysis is limited to thin-walled and beam-like structures with I-, H-, and C-shaped cross-sections. The study further focused on bifurcation analysis, which is not the subject of this paper. Wagner and Gruttmann (2001) applied the finite element method to the Saint Venant torsional problem approximating the warping function and analysing different cross-sections (triangular, rectangular, circular, H-beam and bridge transition profile) of a prismatic and isotropic bar. The study,

however, deals exclusively with out-of-plane deformations. Gonçalves et al. (2010, 2011) introduced a formulation for thin-walled beams that models the in- and out-of-plane deformation with shape functions. Kirchhoff's assumptions, among others, were used in this formulation. Prismatic and isotropic thin-walled structures were investigated and bifurcation analysis performed. Damkilde and Lund (2009) has shown that applying in-plane pressure as external loads to a 2D beam structure with a rectangular cross-section leads to the same cross-sectional deformation and

stresses as a non-linear 3D FE simulation of a prismatic beam, which can be used for structural design optimisation. Carrera et al. (2020) and Carrera et al. (2021) investigated cross-sectional deformations on prismatic structures with different cross-sections caused by large deformations using the Carrera unified formulation (CUF), which is a hierarchical formulation. Both studies were limited to isotropic linear elastic material behaviour. Varello et al. (2013) focused on the cross-sectional deformation of aircraft wings with constant chord length utilising the CUF-

based approach. The results were compared with FE simulations and showed good accuracy with a decrease in computational cost.

Deformations of rotor blade cross-sections have also been investigated, and the importance for blade design was shown in a variety of publications. In this context, Cecchini and Weaver (2005) introduced an energy-based method for thin walled, symmetric cross-sections under bending separating longitudinal and in-plane effects. Herein, the





deformations resulted from crushing forces applied to symmetric cross-sections. Thin cross-sections showed good accuracy compared with FE results. Eder and Bitsche (2015) presented an analysis of asymmetric airfoils under bi-axial bending and discussed the negative impact of the cross-sectional deformation on the fatigue life of the adhesive joints. However, the aforementioned publications did not investigate the aero-elastic impact that cross-sectional deformations may result in.

Preliminary work on the quantification of in-plane cross-sectional deformations of rotor blades was presented in Gebauer and Balzani (2023) and Balzani and Gebauer (2023). Therein, a three-dimensional finite element model was used to calculate the deformed blade shape based on the bending moment distributions from aero-servo-elastic simulations. The deformed positions of nodes associated with a cross-section of interest were then projected onto a plane that was considered the cross-sectional plane in the deformed configuration, and the deformed cross-sectional

shape was obtained. To the best knowledge of the authors, the impact of cross-sectional deformations on the aerodynamic behaviour of the airfoils and the coupling with the aero-elastic response of the blades was not yet investigated. Hence, the main objective of this paper is to study the influence of cross-sectional deformations in the rotor blade on the aero-elastic behaviour of the wind turbine. A simple test case with respect to normal operation of the wind turbine and a constant wind field is selected for the initial investigation presented in this paper. The aim is to provide

a first quantification of the aero-structural coupling due to cross-sectional deformations.

The content of the paper is organized in four sections. The workflow and methods are described in section 2. In section 3, the cross-sectional deformations are analysed for a reference turbine, and the changes in chord length and lift and drag coefficients are presented. The impact of the geometrical changes on the aero-elastic turbine response are examined and discussed in section 4. Section 5 draws conclusions and gives an outlook on future work.

## 2  Methods

The study is conducted on the model of the IEA 15 MW reference wind turbine (RWT). The rotor blades have a length of 117 m. Data of the wind turbine for load simulations and of the blades to create a 3D finite element (FE) model is provided in Gaertner et al. (2020). To analyse the influence of the cross-sectional deformations of the rotor blade on the wind turbine behaviour, a two-stage process is applied. A flow chart of the process is given in Fig. 1.

First, a 3D FE model of the rotor blade was created to obtain the 3D blade geometry. The in-house tool MoCA (Model Creation and Analysis tool, see Noever-Castelos et al. (2022)) was used for this purpose, because the parameterisation is very similar to the available blade model data. The longitudinal interpolation of the cross-sections embedded in MoCA resulted in differences in the relative thickness distribution compared to the reference model data, especially in the inboard region. It was thus important to model the 3D geometry with MoCA already at this

stage and to modify the reference model accordingly, as MoCA was also used in subsequent steps. Not modifying the blade geometry would thus have resulted in inconsistent comparisons at a later stage. Based on the FE model geometry, the aerodynamic polars were calculated for each nodal position along the blade. The polars were fed back into

**Figure 1.** Flowchart of the routine to calculate the aerodynamic polars considering the cross-sectional deformations and include them in the aero-servo-elastic simulation for the wind turbine.



the turbine simulation model and were used to calculate the aerodynamic loads via the blade element momentum theory (Jonkman et al., 2015). For the load calculation, an aero-servo-elastic simulation of the turbine was performed using OpenFAST (National Renewable Energy Laboratory, 2023). From here, the generated loads were transformed from the OpenFAST coordinate system into a global coordinate system that is used in the 3D FE simulation. With the FE model and the transformed loads, a simulation was started in Ansys (Ansys®, 2020) assuming clamped boundary conditions at the blade root. Following, the deformed 2D cross-sections were extracted from the resulting 3D blade model in the deformed configuration. For the procedure to extract the deformed cross-sections the reader is referred to Gebauer and Balzani (2023) and Balzani and Gebauer (2023). The deformed cross-sections were used to re-calculate the lift and drag coefficients. The aerodynamic polars were re-imported into the wind turbine simulation model, the aero-servo-elastic simulation was repeated and the results were analysed with respect to a change in loads and turbine behaviour. The individual steps are described in the following sections in more detail.

## 2.1 Three-dimensional mechanical behaviour of the blade

In order to determine the impact of cross-sectional deformations on aerodynamics and the wind turbine behaviour, the rotor blade must be modelled with sufficient accuracy. Moreover, the blade model must allow for the extraction of deformed shapes of the cross-sections, as this information is needed for a subsequent turbine simulation. A 3D finite shell element model was thus employed, as such model provides a reasonable compromise between accuracy and computation time. The FE solver was Ansys® (2020). The blade model was created with the in-house Model Creation and Analysis tool MoCA that had been validated earlier using a physical full-scale blade test (Noever-Castelos et al., 2022) and had thus been proven to provide a sufficient level of accuracy.

To verify the FE blade model, it is compared with the beam model of the reference turbine. Figure 2 shows selected geometrical (top) and structural (bottom) data as functions along the blade span for both the reference model and the MoCA model. The geometrical data comprise the chord length (top left) and the relative thickness (top right), the structural data the second moments of inertia in the flapwise (out-of-plane bending, bottom left) and the edgewise (in-plane bending, bottom right) direction. For the MoCA model, the second moments of inertia were calculated using BECAS (Blasques and Stolpe, 2012), which is a 2D FEM tool that provides cross-sectional stiffness and mass matrices based on a 3D FE model and is widely used in the research community and the wind energy industry. The geometric parameters were chosen because they have a major influence on the aerodynamic behaviour of the rotor. The second moments of inertia were selected as structural representatives, because the subsequent investigations focus on the bending behaviour of the blades. In all subplots of Fig. 2, the data from the MoCA model is displayed by cross markers, that of the IEA 15 MW RWT model by circle markers. The relative deviation for all parameters are presented in grey, with the relative deviation $\epsilon$ being defined by the expression

$$\epsilon = \frac{x_{\mathrm{gen}} - x_{\mathrm{ref}}}{x_{\mathrm{ref}}} \ . \tag{1}$$





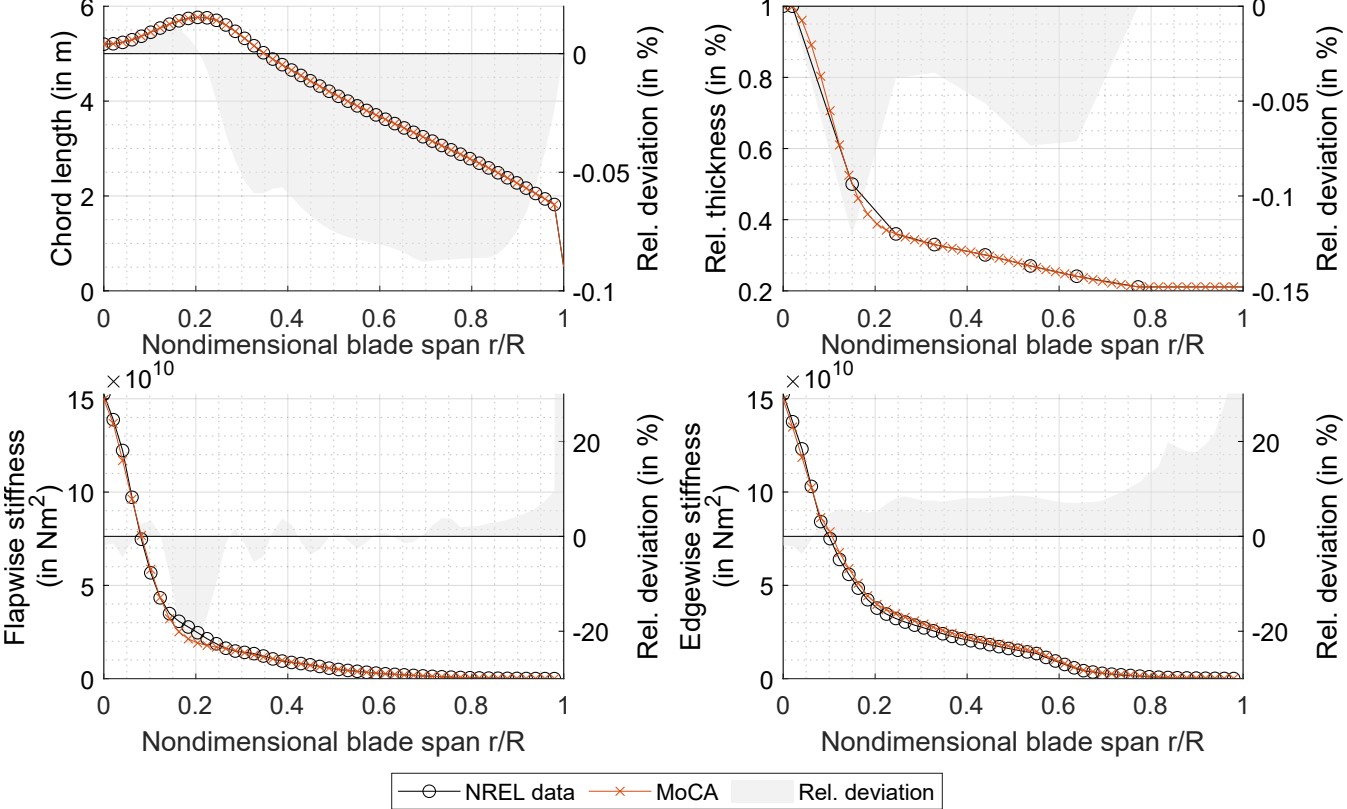

**Figure 2.** Comparison of the MoCA-based FE model and the IEA 15 MW reference turbine model. The chord length (top left), the relative thickness (top right), and the flapwise (bottom left) and edgewise (bottom right) area moments of inertia are plotted as functions of the normalised blade span. The differences between the models are presented as relative deviation.

Herein, $x_{\mathrm{ref}}$ is a reference data point from the reference turbine model and $x_{\mathrm{gen}}$ is a data point generated by MoCA. In the following, the relative deviation is defined accordingly. The relative deviation is only evaluated at spanwise positions where data points are available for both the reference and the FE model.

Figure 2 shows that the relative deviation of the chord length is below -0.1 %. This parameter is therefore considered to be accurately mapped. The relative thickness is also well represented in the FE model at spanwise 130 positions where data points are given in the NREL data of the reference model (relative deviation below -0.12 %). However, in the reference model, the resolution of data points is relatively coarse, which are only given at the positions where the basic airfoils are defined. In the FE model, data points are also calculated equidistantly in between using piecewise cubic interpolation splines. Especially in the region where the geometry has large gradients, i. e., at spanwise positions of $r/R < 0.25$, this results in significant deviations compared to a linear interpolation in 135 the reference model plotted in Fig. 2.



The MoCA-based cross-sections of the FE model were exported to BECAS (Blasques and Stolpe, 2012), which is a 2D FE tool that served to calculate the cross-sectional stiffnesses. The resulting flapwise and edgewise second moments of inertia (bending stiffnesses) are plotted exemplarily together with the NREL reference data at the bottom of Fig. 2. The stiffnesses generally agree very well. The deviation is largely less than -10 %. However, it

can be seen that there is a significant deviation in the flapwise bending stiffness of up to -22 % in the range of the maximum chord length (around $r/R = 0.2$). This is due to the difference in relative thickness, which defines the distance between the spar caps and thus has a major impact on the flapwise bending stiffness. A significant difference in both the flapwise and the edgewise stiffnesses occurs at the blade tip due to a difference in chord length at the tip. However, from a structural point of view, the blade tip is irrelevant, because the free end of the blade is

unloaded. Moreover, the relative deviation increases towards the tip, because the absolute stiffness values decrease and approach zero towards the tip. The general small deviation of stiffnesses is likely due to differences in the tool chain to extract stiffnesses from a 3D model.

The MoCA model was used to calculate the cross-sectional deformations at a later stage of this study. For a consistent comparison of the aerodynamic behaviour of the undeformed and the deformed blade and due to the

differences of the structural behaviour between the reference model and the MoCA-based model, the MoCA-based beam model was used in the following and the blade data for the OpenFAST model was modified accordingly. This includes the relative thickness distribution, i. e., the aerodynamic description of the blade.

A mesh convergence study with respect to natural frequencies was carried out for the 3D FE model. The converged mesh consisted of 128,986 nodes and 131,008 quadrilateral, 4-noded shell elements with linear shape functions and

6 nodal degrees of freedom (element type SHELL181 in Ansys (Ansys®, 2020)).

For the geometrically non-linear FE simulations, the rotor blade was fully clamped at the blade root. The loads were calculated via an aero-servo-elastic turbine simulation with OpenFAST (see section 2.3). Concentrated forces in out-of-plane and in-plane direction of the rotor were applied to the 3D model at discrete locations along the blade span. The magnitude of the forces were calibrated so that the flapwise and edgewise bending moments from the loads

simulations were well approximated. Since the ElastoDyn module was used in the OpenFAST simulation, torsion was not accounted for. Hence, torsion loads were neglected in the 3D model and the loading was bending-dominated. Applying flapwise and edgewise bending moments simultaneously results in a static and multi-axial load scenario.

The loads were applied via multi point constraints that represented a load introduction similar to load frames used in physical full-scale blade testing (IEC61400-23, 2001). As described in Noever-Castelos et al. (2022) an extra

cross-section is introduced at the load frame position. Here, the master node is defined at the centre of gravity of this cross-section. The substitute loads then act on the load frame position in the respective shear center, that was calculated with BECAS (Blasques and Stolpe, 2012). The loads affect the deformations in the spanwise vicinity of the load frames. According to IEC61400-23 (2001), a region corresponding to the chord length can not be taken into account in spanwise direction on both sides of the load frames (Saint-Venant principle). At the same time, the

bending moment must be mapped as accurately as possible. For the positioning of the load frames the blade was



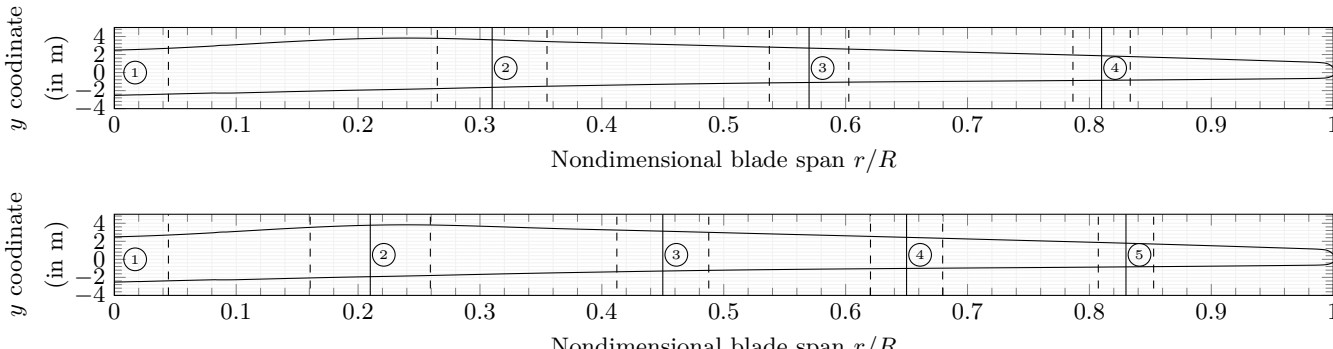

**Figure 3.** Load frame positions. Solid lines represent the radial load frame positions in the FE model. Dashed lines indicate the areas that cannot be evaluated.

divided into 50 equidistant points. An investigation was carried out for two, three, four and five load frames in which the effect of spanwise positions on the utilisable blade length and the error in bending moment were calculated. In all cases, the clamping at the blade root was considered the first load frame.

Based on the results of that investigation, five load frames were selected and positioned along the blade, see the
bottom image of Fig. 3. The cross-sectional deformations were only evaluated between the load frame regions marked by dashed vertical lines. To be able to also analyse the areas around the load frames, a second set of four load frames shifted in spanwise direction were used, see the top plot of Fig. 3. Hence, for each load case two simulations were performed with four and five load frames, respectively. The outermost blade tip was not considered as a possible load frame position, as 3D FE simulations have shown that there are stability issues otherwise. That is acceptable,
because the outermost region of the blade is not subjected to high mechanical loads. Hence, high cross-sectional deformations are not expected there.

## 2.2 Airfoil polars

For the deformed cross-sections, the airfoil polars needed to be calculated. For a consistent comparison, the polars of the undeformed cross-sections had to be calculated with the same methods. The panel method implemented in
XFOIL (Drela, 1989) was used for this purpose. It is known that XFOIL predicts the linear range of the lift coefficient well (Lennie et al., 2015) for thin airfoils, i. e., the results are good for small angles of attack. In the analysed test case, the turbine operation mode mostly revealed angles of attack of less than 10°, which were only exceeded in the blade root area. High-fidelity computational fluid dynamics (CFD) was used to compare the polars and ensure the validity of XFOIL results.
Utilizing the open-source CFD solver OpenFOAM v2012, a three-dimensional transient Unsteady Reynolds-Averaged Navier-Stokes (URANS) simulation was conducted to determine the static polars of two selected deformed airfoils at spanwise positions of $(r/R)_A = 0.21$ and $(r/R)_B = 0.75$. The simulation was executed for a Reynolds number



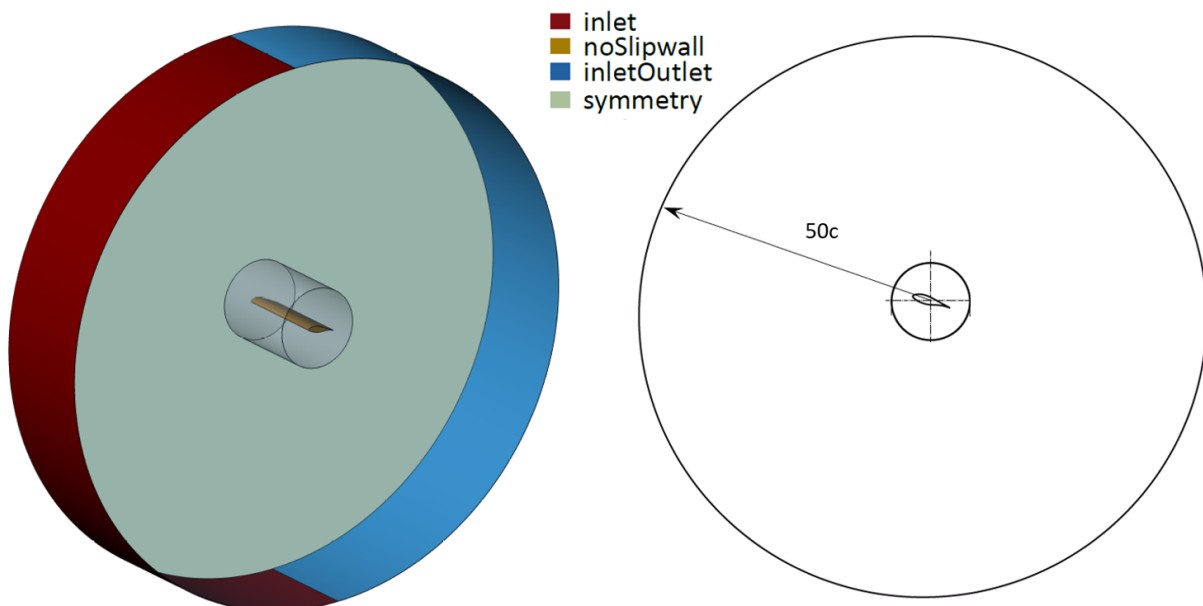

**Figure 4.** Boundary conditions and dimensions of the computational domain used to simulate the static polars with URANS.

of $Re = 3 \times 10^6$. The primary objective was to anticipate flow separation on the suction side of the airfoil, which impacts the lift coefficient at increasing angles of attack. The computational domain, illustrated in Fig. 4, takes the

form of a cylinder with a diameter of $50c$ and a length of $1c$ in the span-wise direction, where $c$ is the chord length. These dimensions, previously employed by Yalcin et al. (2021), have proven effective in obtaining simulation results independent of far-field and spanwise mesh resolution. This allows for an in-depth analysis of flow phenomena around the airfoil. Boundary conditions for the computational domain are outlined in Tab. 1. Specifically, at the inlet, a uniform velocity and a zero gradient for kinematic pressure are applied, while at the outlet, a zero gradient for the

velocity and a uniform kinematic pressure of $p/\rho = 0$ (indicating incompressible flow) are enforced. Wall boundaries adopt a symmetry condition, except for the airfoil surface where a zero-velocity condition prevails. Initial values for the internal fields are estimated using the inlet velocity, the hydraulic diameter, and the Turbulence Intensity (TI = 0.1%). The simulation employs the transient and incompressible pimpleFoam solver, incorporating the PIMPLE (merged PISO-SIMPLE) algorithm for pressure and velocity coupling (Issa, 1986). OpenFOAM utilizes the finite

volume (fV) method to discretize differential terms in the RANS equation. The discretisation of the governing system of equations relies on a second-order finite volume approach in both space and time. Spatial discretisation employs a second-order upwind method as outlined by Warming and Beam (1976), while temporal discretisation follows the implicit three-point backward Euler scheme. Turbulence-related additional terms are modeled using the Shear Stress





**Table 1.** Initial and boundary conditions for the URANS simulations.

| Boundary | $U_\infty, [\text{m/s}]$ | $p/\rho, [\text{m}^2/\text{s}^2]$ | $k, [\text{m}^2/\text{s}^2]$ | $\omega, [1/\text{s}^2]$ | $nut, [\text{m}^2/\text{s}]$ |
|---|---|---|---|---|---|
| Inlet | fixedValue, $U_\infty$ | zeroGradient | $I_\text{t} = 0.05$ | $L_\text{mixing} = 0.2625$ | calculated |
| Outlet | zeroGradient | fixedValue, 0 | zeroGradient | zeroGradient | calculated |
| Airfoil | fixedValue, 0 | zeroGradient | fixedValue, $1e-9$ | omegaWallFunction | nutlowReWallFunction |
| Walls [sides] | symmetry | symmetry | symmetry | symmetry | symmetry |

**Table 2.** Numerical schemes for the URANS simulations.

| Operator | Selected schemes | Accuracy order |
|---|---|---|
| ddtSchemes | backward | 2$^\text{nd}$ |
| gradSchemes ($\nabla$) | Gauss linear | 2$^\text{nd}$ |
| divSchemes ($\nabla \cdot$) | Gauss linearUpwind | 2$^\text{nd}$ upwind |
| laplacianSchemes ($\nabla^2$) | Gauss linear limited corrected, 0.5 | 2$^\text{nd}$ |

Transport (SST) approach, as introduced by Menter (1994). A summary of the applied numerical schemes can be
found in Tab. 2.

Statistical convergence was monitored using the methodology introduced by Ries et al. (2018). The maximum
ensemble error for all simulations was $1.8\text{e}^{-2}$. The non-dimensional cell-height

$$y^+ = \frac{y \cdot \sqrt{\tau_\text{w}/\rho}}{\nu} \approx 1 \tag{2}$$

during all time steps, and $x^+_\text{max} = 381$. These cell dimensions are sufficient to investigate integral blade loads for
attached and separated flow (Ahrens et al., 2022).

The results of the URANS simulations are compared with the XFOIL results in Fig. 5. The lift coefficients agree
very well. We can thus conclude that XFOIL can be used for the calculation of the polars along the blade. Especially
in the outboard region of the blade, see on the right of Fig. 5, the XFOIL results are very accurate for an angle of
attack up to 10° and almost coincide with the URANS results. For the thicker airfoils more close to the blade root,
see on the left of Fig. 5. However, due to the substantially lower computational cost, XFOIL was also used there for
the analysis of the aero-elastic impact of cross-sectional deformations.

## 2.3 Aero-servo-elastic response of the wind turbine

Aero-servo-elastic simulations were carried out to determine the overall dynamic response of the wind turbine and to
calculate the loads that the rotor blades are exposed to. The turbine simulations were carried out in two stages. The
first simulation was executed to observe the behaviour of the system with undeformed cross-sections in the blade.





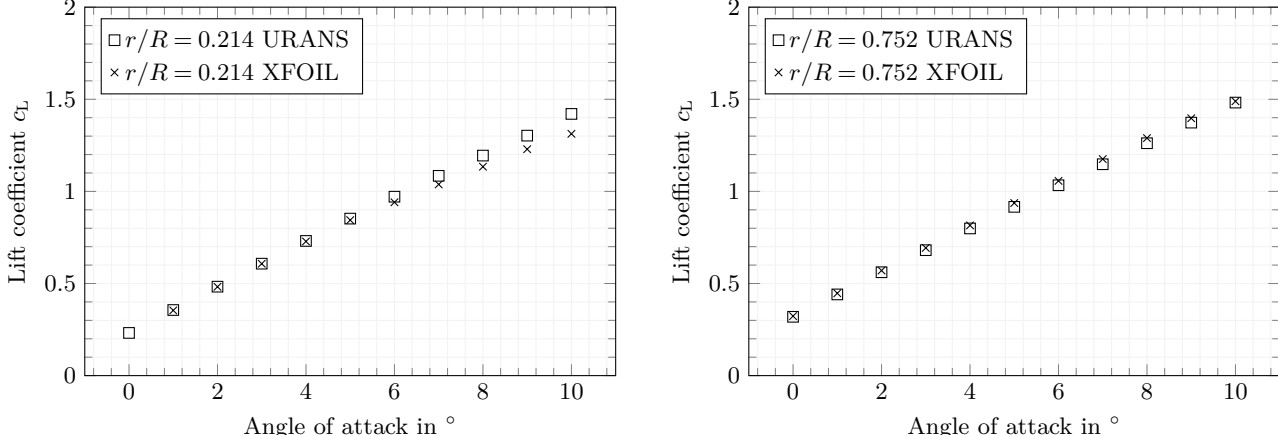

**Figure 5.** Lift coefficient calculated with URANS and XFOIL. The maximum sampling error is $1.84\mathrm{e}^{-2}$ and was calculated according to Ries et al. (2018).

The second simulation was carried out for deformed cross-sections, so that the dynamic behaviour of the turbine with and without cross-sectional deformations could be compared.

The aero-servo-elastic simulations were performed with OpenFAST (National Renewable Energy Laboratory, 2023). ElastoDyn (National Renewable Energy Laboratory, 2024) was used to model the structural dynamics of the

blades. Therein, an Euler-Bernoulli beam theory in combination with a modal reduction is employed, where the first two flap- and the first edgewise bending modes are considered.

To focus on the influence of the cross-sectional deformations on the aero-elastic behavior of the turbine, a constant wind field at rated wind speed of 10.5 m/s was chosen in this investigation. At this wind speed the rotor experiences the highest thrust force. Moreover, this is the point at which the rotor blade is not yet pitched, so that flap- and

edgewise directions coincide with the out-of-plane and in-plane directions with respect to the rotor plane.

The simulation was carried out until the rotor blade behaviour became periodic. A full periodic rotation of one rotor blade was analysed, and the blade positions with an azimuth angle $\beta \in \{0°, 90°, 180°, 270°\}$ were the basis for the subsequent investigations, see also the blade highlighted in blue in Fig. 6. A simulation was carried out to extract the reference loads along the blade for the aforementioned azimuth positions. The flapwise and edgewise

bending moments were applied to the 3D FE model, with which the deformed cross-section shapes were calculated. The new polars were determined with XFOIL and were fed into additional aero-servo-elastic simulations. Hence, five aero-servo-elastic simulations were carried out, one with undeformed cross-sections, and four with deformed cross-sections calculated from the bending moment distributions at the four azimuth positions mentioned above and highlighted in Fig. 6.





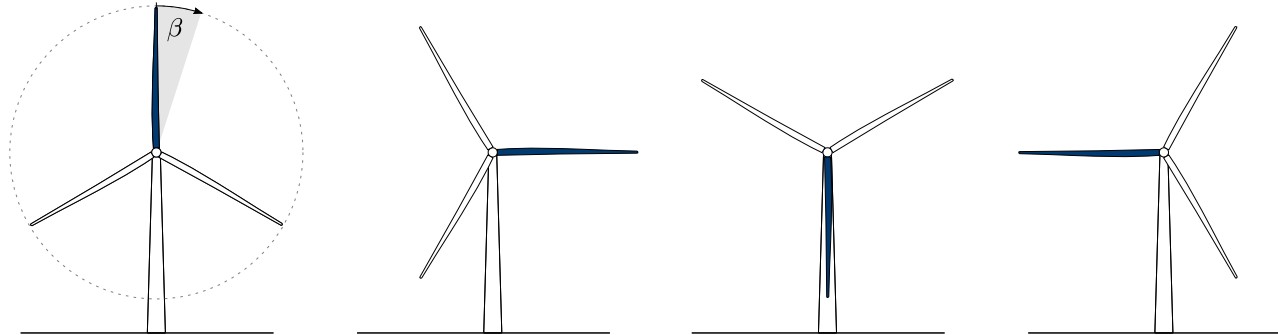

**Figure 6.** Wind turbine rotor positions considered in this study. The rotor blade loads are evaluated at four azimuth angles $\beta$ indicated by the blue rotor blade, i. e., $\beta = 0°$, $\beta = 90°$, $\beta = 180°$, and $\beta = 270°$.

## 3 Cross-sectional deformations

In this section, the cross-sectional deformations at two positions along the blade are presented and discussed.

Recall the blade element theory (Hau, 2013). Each rotor blade is divided into so-called blade elements. The lift and drag forces on blade element level, which are denoted by $\mathrm{d}F_\mathrm{L}$ and $\mathrm{d}F_\mathrm{D}$, can be calculated at steady-state conditions by the relations

$$\mathrm{d}F_\mathrm{L} = \frac{\rho}{2}\, c_\mathrm{L}(\alpha)\, u_\mathrm{rel}^2\, c\, \mathrm{d}r \quad \text{and} \quad \mathrm{d}F_\mathrm{D} = \frac{\rho}{2}\, c_\mathrm{D}(\alpha)\, u_\mathrm{rel}^2\, c\, \mathrm{d}r \,, \tag{3}$$

where $\rho$ denotes the mass density of air, $c_\mathrm{L}(\alpha)$ and $c_\mathrm{D}(\alpha)$ are the lift and drag coefficients at a given angle of attack $\alpha$, $u_\mathrm{rel}$ is the relative inflow velocity, $c$ is the chord length, and $\mathrm{d}r$ is the radial extension of the blade element under consideration. The change in airfoil geometry due to deformation can affect the chord length $c$ as well as the lift and drag coefficients and therefore have an impact on the lift and drag forces.

### 3.1 Change in airfoil shape

Aero-servo-elastic simulations with a duration of 600 s were carried out in OpenFAST with the IEA 15 MW RWT and a constant wind field at the rated wind speed. The internal loads at the four rotor positions (blade at 3, 6, 9 and 12 o'clock) were extracted. These bending moments were transformed into the global blade coordinate system, which was required for the 3D FE simulations. The resulting bending moments are shown together with the blade positions and the cross-sectional deformations in Figs. 7 to 10. The dashed lines represent the flapwise bending moment while the dash-dotted lines the edgewise bending moment, respectively. In each blade position the absolute values of the bending moments are highest at the blade root and continuously decrease to zero at the blade tip.

Fig. 7 shows the loads and cross-sectional deformation for the blade position of $\beta = 0°$ (12 o'clock). The rotor blade points upwards, so that its weight has a negligible effect on the bending moments. Both the flapwise and the edgewise bending moments have a positive sign. This means that the blade bends towards the suction side and





towards the leading edge. The maximum flapwise bending moment is approximately 10.6 times bigger than the maximum edgewise bending moment.

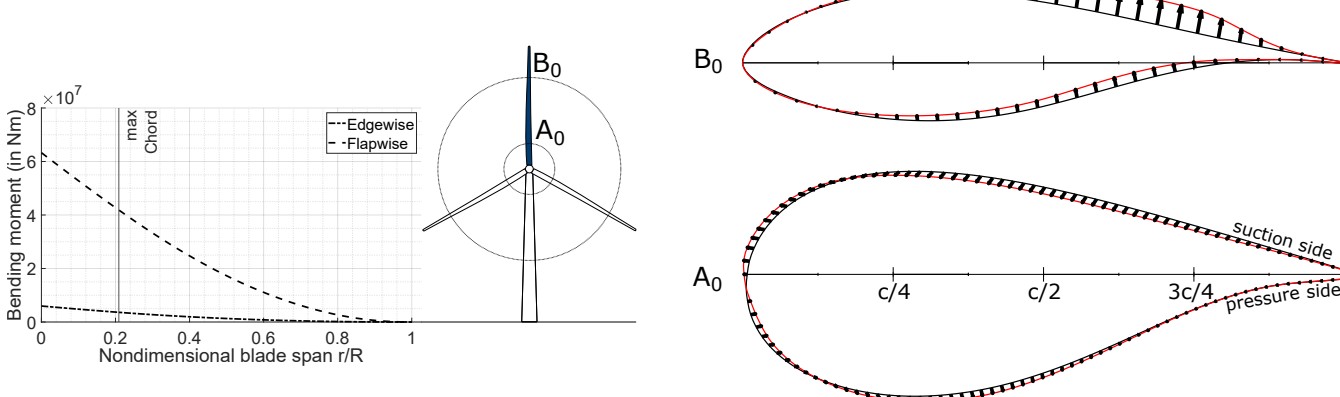

**Figure 7.** Bending moment distributions and cross-sectional deformations at two positions along the rotor blade for $\beta = 0°$ (12 o'clock) azimuth position of the blade. The rotor positions are $(r/R)_A = 0.21$ and $(r/R)_B = 0.75$. The cross-sectional deformation was plotted with a scaling factor of 5.

As shown in Fig. 8, the tip of the rotor blade points to the side at $\beta = 90°$ (3 o'clock), with the leading edge downwards. In this position, the weight of the blade has a major influence on the edgewise bending moment. Hence, the edgewise bending moment is bigger than at $\beta = 0°$. Since the wind field is constant, the flapwise bending moment remains almost the same. The maximum flapwise bending moment is still approximately 2.6 times bigger than the edgewise bending moment.

At a blade position of $\beta = 180°$ (6 o'clock), the rotor blade is again vertically aligned and points downwards. Fig. 9 shows this rotor position. The bending moments are similar to those at a blade position of $\beta = 0°$, because the blade's weight is negligible and the bending moments are governed by aerodynamic forces.

At a blade position of $\beta = 270°$ (9 o'clock), the blade tip points to the side with the leading edge upwards. The weight is now counteracting the circumferential aerodynamic forces, so that the edgewise bending moments become negative. The absolute values of the flapwise bending moment at the blade root is 5.3 times bigger compared to the edgewise bending moment.

The bending moments were each applied to the 3D FE blade model and the respective simulations were carried out. The deformed 2D cross-sections were extracted from the resulting deformed rotor blade. For the four blade positions, two cross-sections were exemplarily chosen for investigating the cross-sectional deformations in detail. Cross-section A is close to the maximum chord position at $(r/R)_A = 0.21$ and cross-section B in the outboard region of the blade at $(r/R)_B = 0.75$. The cross-sectional deformations of these two cross-sections are shown on the right-hand side of Figs. 7 to 10. For visualisation purposes, the chord lengths of the cross-sections are normalized and the deformation vectors are magnified by a scaling factor of 5. The shear webs are not shown, since they are not important for the



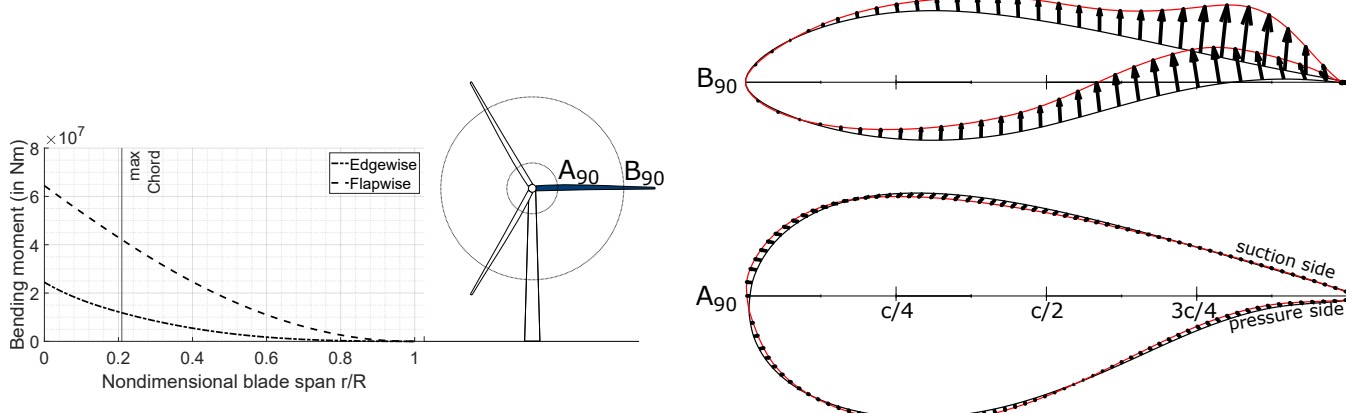

**Figure 8.** Bending moment distributions and cross-sectional deformations at two positions along the rotor blade for $\beta = 90°$ (3 o'clock) azimuth position of the blade. The rotor positions are $(r/R)_A = 0.21$ and $(r/R)_B = 0.75$. The cross-sectional deformation was plotted with a scaling factor of 5.

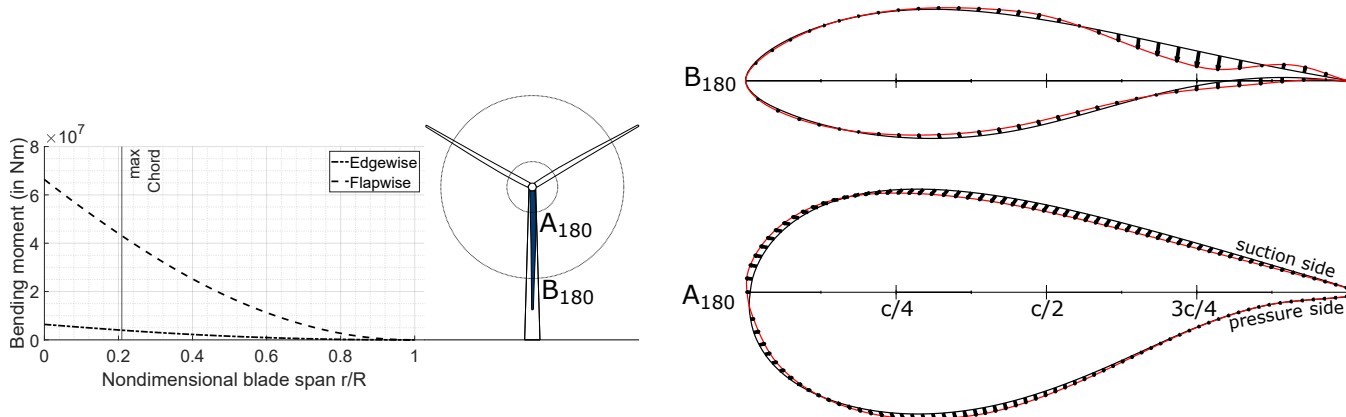

**Figure 9.** Bending moment distributions and cross-sectional deformations at two positions along the rotor blade for $\beta = 180°$ (6 o'clock) azimuth position of the blade. The rotor positions are $(r/R)_A = 0.21$ and $(r/R)_B = 0.75$. The cross-sectional deformation was plotted with a scaling factor of 5.





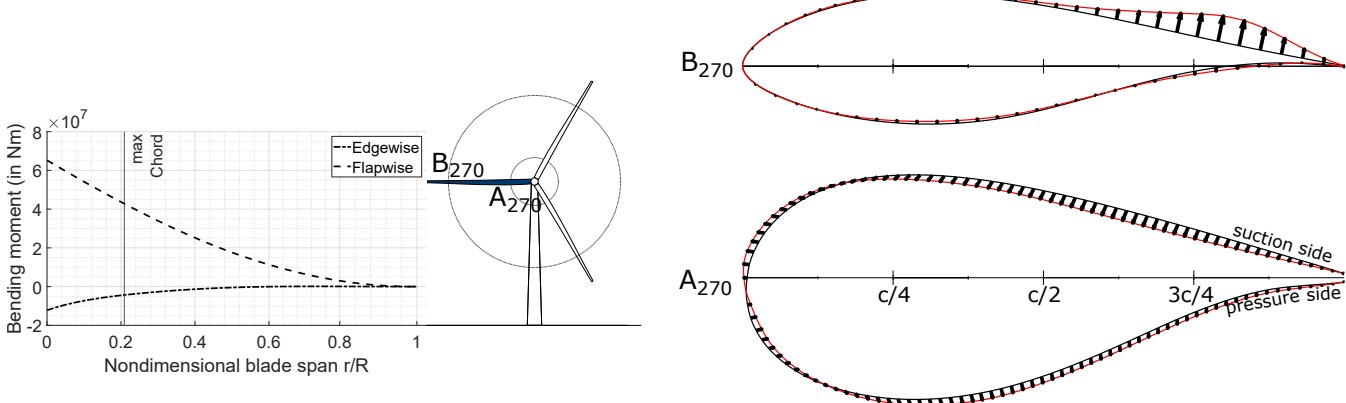

**Figure 10.** Bending moment distributions and cross-sectional deformations at two positions along the rotor blade for $\beta = 270°$ (9 o'clock) azimuth position of the blade. The rotor positions are $(r/R)_A = 0.21$ and $(r/R)_B = 0.75$. The cross-sectional deformation was plotted with a scaling factor of 5.

aerodynamic forces. The black lines show the undeformed geometries of the cross-sections, while the red lines show the deformed cross-sections. In between, arrows are used to highlight the deflection vectors.

For the cross-section A at $\beta = 0°$ (denoted by $A_0$), small deformations are detectable. On the suction side between
the trailing edge and approximately $c/4$, the shell deforms into the cross-section. From approximately $c/4$ up to the leading edge, the suction side shell deforms out of the cross-section. At the trailing edge of the suction side the shell deforms inwards the cross-section. From the leading edge up to approximately $c/8$, the shell on the pressure side deforms into the cross-section. From there on up to approximately $5c/8$, the pressure side shell deforms out of the cross-section. After this point up to the trailing edge, the pressure side shell hardly deforms, neither out of
nor into the cross-section. The deformation of cross-section A at $\beta = 180°$ (denoted by $A_{180}$) looks very similar. At $\beta = 90°$, the trailing edge of cross-section A (denoted by $A_{90}$) rotates slightly clockwise, i. e., the deformation into the cross-section on the suction side vanishes and the zero deformation on the pressure side changes to a deformation into the cross-section. The deformation in the rest of the cross-section is very similar to $A_0$ and $A_{180}$. At $\beta = 270°$, the trailing edge slightly rotates counter-clockwise, i. e., there is a deformation into the cross-section on the suction
side and a deformation out of the cross-section on the pressure side. The sign of the edgewise bending moment is a major difference in loading between $A_{90}$ and $A_{270}$, and the magnitude of the edgewise bending moment is changing significantly during one rotation. Contrarily, the flapwise bending moment is almost constant. The deformation in the vicinity of the trailing edge changes with the rotation, but in the other regions of the cross-section the deformation is constant. Hence, the deformation in the vicinity of the trailing edge seems to be governed by the edgewise bending
moment (rotation of the trailing edge clockwise for positive edgewise bending moments and rotation of the trailing edge counter-clockwise for negative edgewise bending moments). The rest of the deformation is similar in all positions and is thus dominated by flapwise bending.





The cross-section B does not show any deformation around the leading edge at all four positions of the blade. At $\beta = 0°$ (denoted by $B_0$), the deformation at the trailing edge is also almost zero. However, in the trailing edge shell

of the suction side, there is a significant deformation out of the cross-section, and on the pressure side into the cross-section in a long region between the shear webs and the trailing edge. The deformation at $\beta = 90°$ (denoted by $B_{90}$) shows an additional rotation clockwise, which is similar to the findings in cross-section A. Hence, the deformation out of the cross-section on the suction side and into the cross-section on the pressure side are more pronounced compared to $B_0$. At $\beta = 180°$ (denoted by $B_{180}$, the deformation shows a clear difference compared to $B_0$. It appears

as if there was a buckle into the cross-section close to the trailing edge on the suction side and a small buckle out of the cross-section on the pressure side. The deformation at $\beta = 270°$ (denoted by $B_{270}$) looks qualitatively similar to $B_{90}$, but the amplitude is much smaller. The dependencies between the cross-sectional deformations and the loading situations associated with the blade positions are not as clear for cross-section B as for cross-section $A$.

The cross-sectional deformations were determined for all cross-sections along the blade. However, due to space

limitations, they cannot be discussed in detail here. Nevertheless, the changes in chord length and in lift and drag coefficients are presented in the following for all cross-sections.

## 3.2   Change in chord length

In addition to the actual change in geometry, a change in chord length was also observed. The chord length is defined as the distance between leading and trailing edge and was computed for the deformed cross-sections. The deviation

between the chord length of the undeformed and the deformed cross-sections are plotted in Fig. 11. Each graph represents one blade position. The relative deviation was plotted against the normalised spanwise coordinate $r/R$, where $r$ is the actual spanwise position and $R$ is the spanwise position of the blade tip.

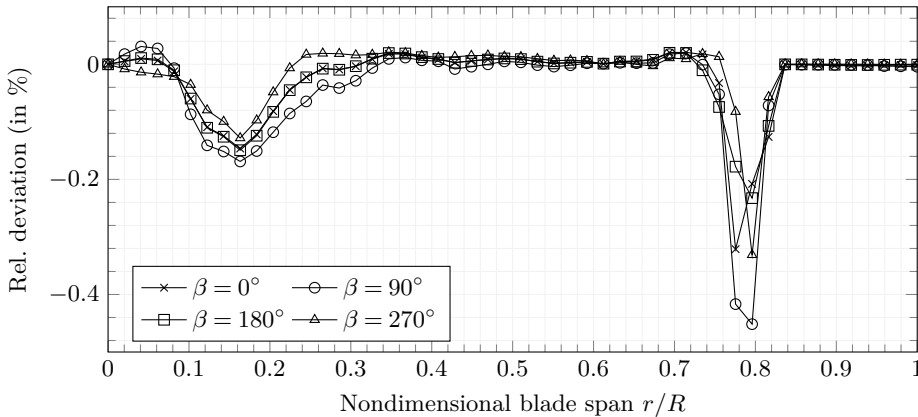

**Figure 11.** Relative deviation of the chord length between the deformed and the undeformed cross-section, plotted as a function of the normalised spanwise position along the blade.





The relative deviations of the chord length show a similar behavior for all four blade positions. There are two regions along the blade with a significant decrease in chord length. The first one is below 30 % of the blade length

with a maximum chord length decrease of -0.16 %. This part of the blade also includes the maximum chord length and is subjected to the highest bending moments. The second portion with even higher relative deviation up to -0.45 % is beyond 70 % blade length up to the point where the last load frame is located. Because there is no load frame beyond 82 % blade length, no bending moments occur in this portion of the blade in the numerical model. The radial portions between 30 % and 70 % blade length show a negligible amount of chord length deviation for the

investigated load cases. The spanwise positions up to 10 % blade length is not evaluated here, because of multiple reasons. First, the deformations of the numerical model are not representative due to the fixed boundary conditions at the blade root. Second, the aerodynamic coefficients of the airfoils used in this section of the blade can not be evaluated with XFOIL, because of too high relative thicknesses. Due to the low radius and the low relative inflow velocity, this area of the blade has only a very small to negligible contribution to the aerodynamic performance and

the aerodynamic loads of the blade, so that the error is considered negligible.

### 3.3 Change in lift and drag coefficients

The deformed cross-sections were used to recalculate the aerodynamic performance of the blade in operation. Therefore, XFOIL simulations were conducted with the new cross-section geometries for the four blade positions. The XFOIL results were included in OpenFAST simulations. Because time dependent changes in airfoils can not be

modeled in OpenFAST, one simulation was conducted for each load scenario of the four blade positions using the same deformation for the entire rotor. The four simulations were then evaluated only at the rotor blade position corresponding to the position where the load was extracted. The lift and drag coefficients, each plotted against the normalised spanwise position along the blade for the four blade azimuth positions, are presented on the left-hand side of Fig. 12. Qualitatively, the distributions of the coefficients is similar for all blade positions. Quantitatively,

there is a general slight difference in lift coefficients depending on the blade position. The quantitative difference in drag coefficients is not significant. In spanwise regions close to the root and close to $r/R = 0.8$, there are irregularities in the aerodynamic coefficients, especially in lift. These are associated with more significant and local cross-sectional deformations in these regions, see also section 3.2.

Note that the angle of attack also changes with different rotor positions due to the tilt and cone angles in the

rotor and the associated vertically inclined inflow. Hence, we need to look at the relative deviations in lift and drag coefficients, which are shown on the right-hand side of Fig. 12. The relative deviation is calculated according to Eq. 1, with the undeformed blade simulation as the reference and the corresponding rotor position simulation as the generated value. Two regions along the blade span can be identified that show a change in aerodynamic properties. The first region is at around 20 % of the span, the second at around 80 % of the blade span. These were also the

regions with the most significant changes in airfoil shape and chord length due to the cross-sectional deformations. A possible explanation is that 20 % blade span corresponds approximately with the region of maximum chord length.





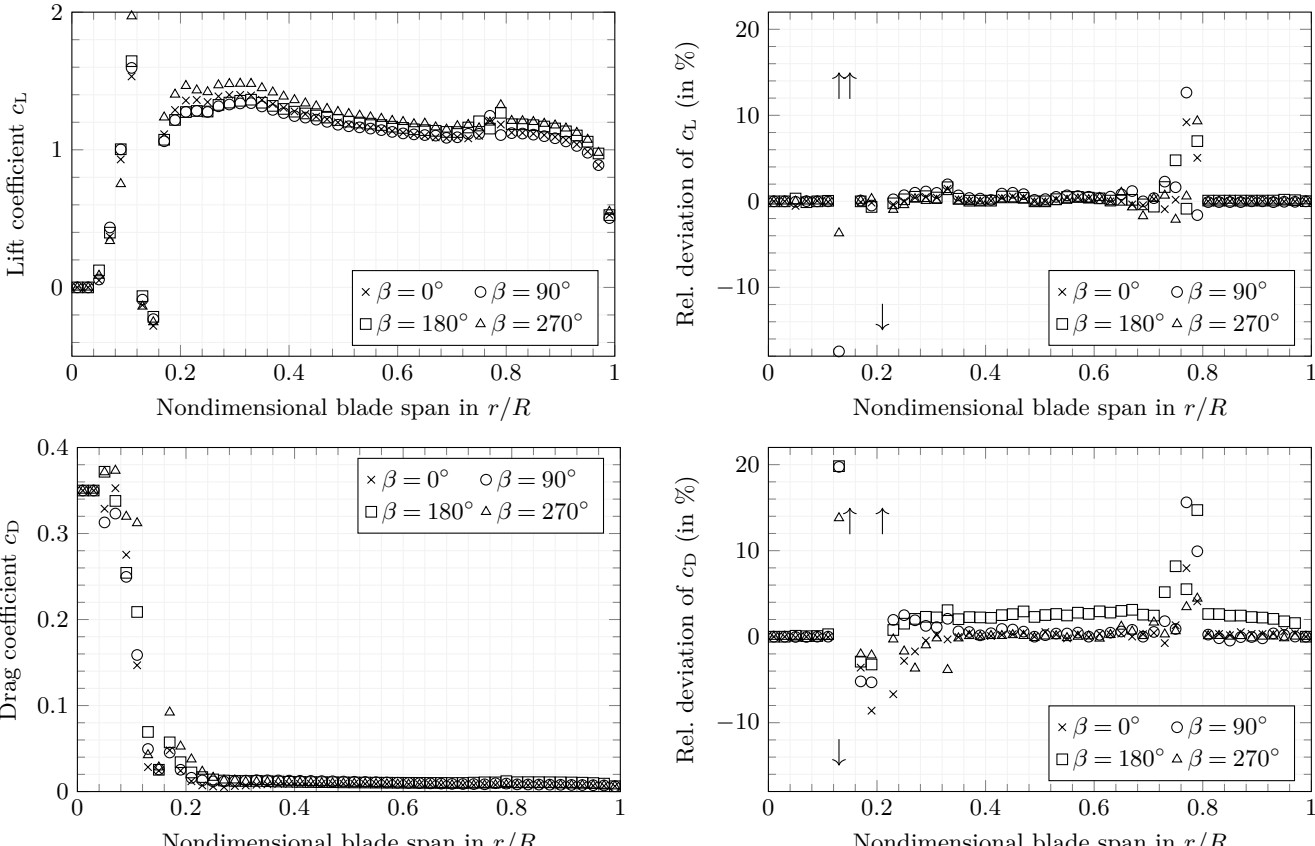

**Figure 12.** Comparison of the aerodynamic performance. The lift coefficient (top left) and the relative deviation of the lift (top right), and the drag coefficient (bottom left) and the relative deviation of the drag (bottom right) are plotted as functions of the normalised spanwise position along the blade for the four analysed azimuth positions of the blade. Values are taken from the respective OpenFAST simulations.

There, the trailing edge panels are the longest panels without stiffeners (shear webs) in the blade. Hence, small relative changes have a big absolute impact. Even though the relative deviation in lift coefficient is the highest here with 146 %, this region close to the blade root does not play a major role for the aerodynamic performance. The deviation close to the tip is likely linked to a lack of stiffness in this area, which is due to the fact that the blade is not in the final design stage, see Gaertner et al. (2020).

## 4 Coupling effect on the turbine behaviour

The aim of this section is to compare the wind turbine behaviour with and without the cross-sectional deformations of the blade. For the comparison, the chord length distribution and the aerodynamic coefficients were updated in





the OpenFAST simulations. For the cross-sectional deformations in each blade azimuth position, an OpenFAST
simulation was carried out in which the new aerodynamic parameters were used for all three blades. The in-plane
and the out-of-plane blade root bending moments for one full rotation of the rotor will be analysed in the following.

The in-plane blade root bending moments are plotted in Fig. 13. The continuous black line represents the con-
figuration with undeformed cross-sections. The values that are compared with those including the cross-sectional
deformations are highlighted by cross markers. Circular markers are used to show the blade root bending moments
based on the cross-sectional deformations calculated for the respective azimuthal position of the blade. The devia-
tions are relatively small. Hence, snippets of the time series are plotted and the relative deviation $\epsilon$ was calculated. At
azimuth positions of $\beta = 0°$ and $\beta = 180°$, the relative deviations are comparably high with $\epsilon = +1.1$ % and $\epsilon = -1.4$
%, respectively. They are at least one order of magnitude smaller at blade positions of $\beta = 90°$ and $\beta = 270°$ with
$\epsilon = -0.16$ % and $\epsilon = +0.08$ %, respectively.

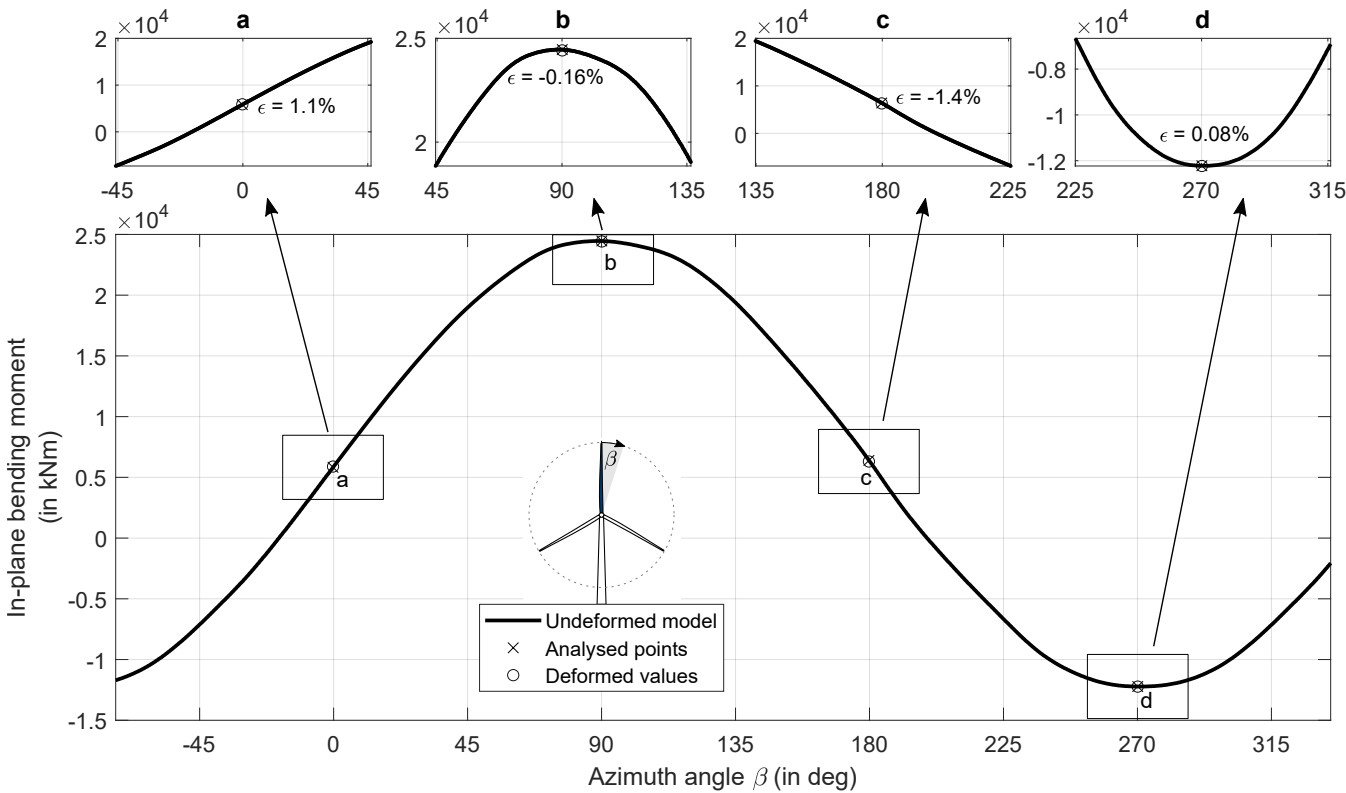

**Figure 13.** Comparison of the in-plane blade root bending moments. The turbine response with cross-sectional deformations
in the four analysed azimuth positions is compared with the turbine response without cross-sectional deformations.

The out-of-plane blade root bending moments are plotted in Fig. 14. The relative deviation is in the same order
of magnitude for all rotor positions and varies between $\epsilon = 0.32$ % at an azimuth position of $\beta = 270°$ and $\epsilon = 0.71$





% at $\beta = 90°$, i.e., in the horizontal blade positions. The relative deviations for the vertical blade positions is approximately the mean of the horizontal positions with $\epsilon = 0.57$ % at $\beta = 0°$ and $\epsilon = 0.48$ % at $\beta = 180°$.

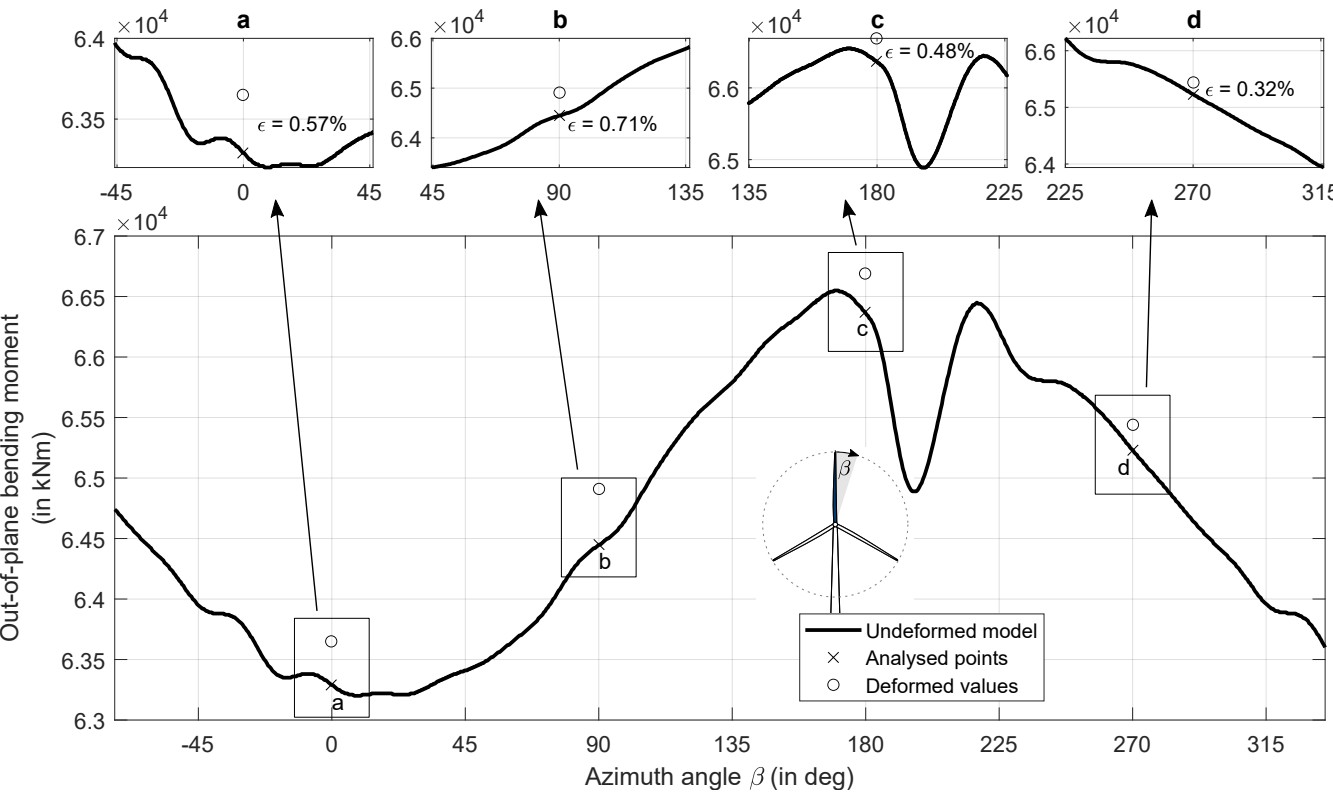

**Figure 14.** Comparison of the out-of-plane blade root bending moments. The turbine response with cross-sectional deformations in the four analysed azimuth positions is compared with the turbine response without cross-sectional deformations.

The impact of cross-sectional deformations on the in-plane bending moments is maximal for vertical blade positions. This is likely due to the asymmetries originating from the tilt and cone angles in the rotor, i.e., the vertically inclined inflow results in the highest relative inflow variations between the top and bottom vertical positions of the rotor. The cross-sectional deformations impact the chord lengths and the aerodynamic coefficients. As the aerodynamic forces and the corresponding bending moments scale quadratically with the relative inflow, it is natural that

the vertical positions reveal the highest impact of cross-sectional deformations. Contrarily, the impact on the out-of-plane bending moments is maximal for the horizontal blade positions. This is natural, because edgewise bending is governed by gravitational forces that are activating maximum bending moments in horizontal positions of the blades.

     This study revealed small changes in lift and drag coefficients, chord length as well as in-plane and out-of-plane

bending moments. Even though the relative deviation of the bending moments is relatively small and potentially





within the uncertainty bandwidth of aero-servo-elastic simulations, the authors would like to emphasise the potential relevance of cross-sectional deformations for very large wind turbine blades. A variety of simplifications entered this initial investigation, i. e., no change in cross-sectional stiffnesses as a result of cross-sectional deformations, a simple load case with constant wind velocity at the rated wind speed, absence of dynamic effects from cross-sectional

deformations (meaning an identical behaviour of all three rotor blades independent of their individual azimuth positions), and the use of the ElastoDyn module for the blades that does not account for torsion or higher bending modes. Even with the aforementioned simplified modeling approaches, deviations regarding the bending moments at the blade root were observed. They are small, but they are finite. Reducing the amount of simplifications may increase the impact of cross-sectional deformations. A comparably low hanging fruit would be to use a BeamDyn model for

the blades in order to more accurately describe the dynamic response of the blades including torsion. It is expected that especially torsion can result in significant in-plane cross-sectional deformations that were neglected in this study, but should be added in future work. Furthermore, other load cases may result in load combinations that result in higher cross-sectional deformations. The trend to larger and increasingly slender blades may also result in more pronounced cross-sectional deformations in turbines exceeding a rated power of 20 MW or more. The aerodynamic

analysis was carried out with a 2D panel method. However, it has not been investigated how the 3D deformation of the blade affects the aerodynamics. Such an investigation would involve a high computational effort, but may be worth to investigate potential radial interactions of cross-sectional deformations. Moreover, the deformation of cross-sections in an operating wind turbine is a dynamic process. Hence, modeling this process dynamically may address other dynamic effects such as dynamic stall, which would be interesting to study, especially in combination

with more complex load cases including turbulence, pitch control, manoeuvres, etc.

## 5 Conclusions and outlook

This paper provides a first study analysing aero-elastic simulations of a wind turbine under consideration of cross-sectional deformations in the rotor blades. To the best knowledge of the authors, such investigation has not been carried out before by other groups.

The wind turbine model under investigation was the IEA 15 MW reference wind turbine. An initial simulation was carried out using aerodynamic coefficients (i.e. lift and drag coefficients) calculated with XFOIL. A simple load case was applied using a constant wind field at the rated wind speed. Four different bending moment distributions based on the azimuth angles of the blade were applied to a detailed 3D FE model of the rotor blade. The resulting cross-sectional deformations were analysed and used for a new calculation of lift and drag coefficients. Additionally,

the change in chord lengths was computed and implemented in the aero-elastic turbine model. The deformations were especially present in the trailing edge panels. The change in chord length was identified in the vicinity of 20 % and 80 % of the blade span. The relative deviation of the chord length in the deformed cross-sections compared with the chord length of the undeformed cross-sections was found to be below $-0.45\%$. Four new OpenFAST simulations





were performed including the new aerodynamic parameters as well as the new chord length distributions that were

calculated for four different azimuth positions of the blade. The results show a maximum change in the blade root

bending moments of -1.4 % for the in-plane bending moment and +0.71 % for the out-of plane bending moment.

These relative deviations are quite small. However, it needs to be considered that a number of simplifications have been made and that a reduction of simplifications can result in a higher impact of cross-sectional deformations on the dynamic behaviour of the turbine. Due to the underlying theory, torsion was not correctly modelled. The

torsional moment was therefore not included in the calculation of cross-sectional deformations, but is considered highly relevant. For a more precise statement regarding cross-sectional deformations, torsion is planned to be included in the near future. Furthermore, the wind field was simplified using the rated wind speed and a constant wind field. The results show also with this assumptions a change. We therefore expect the influence on the aero-elastic simulation to be greater when analysing extreme load cases.

To be able to make a more generalised statement will be the aim for the future.

*Code and data availability.* For legal reasons, code and data cannot be shared at this stage. The authors will try to provide code and data needed for reproduction of the findings by the time of final publication in case the manuscript will be accepted.

*Author contributions.* JG wrote the paper, did the literature research, implemented pre- and postprocessing scripts, carried out the simulations (except CFD), and prepared figures. FP contributed to the calculation of deformed cross-sections on the

basis of 3D FE analyses and internally reviewed the paper. DA conducted CFD analyses and wrote the CFD part of section 2.2. LW supervised DA and internally reviewed the paper. CB provided scientific guidance and supervision to JG in all project phases, internally reviewed the paper, and was responsible for funding acquisition.

*Competing interests.* The authors do not have competing interests.

*Acknowledgements.* The authors gratefully acknowledge the computing time granted by the Resource Allocation Board and

provided on the supercomputer Lise and Emmy at NHR@ZIB and NHR@Göttingen as part of the NHR infrastructure. The calculations for this research were conducted with computing resources under the project nii00172. This research was funded by the Deutsche Forschungsgemeinschaft (DFG, Geman Research Foundation) as part of the Collaborative Research Center 1463 *Integrated Design and Operation Methodology for Offshore Megastructures* (SFB1463 – Project ID 434502799). The authors acknowledge the financial support.





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
