# Peer review of "On the influence of cross-sectional deformations on the aerodynamic performance of wind turbine rotor blades"

_Wind Energy Science, 2024_

## Author Response (AR1)

**RESPONSE TO REVIEW**

Thank you very much to all reviewers and editors for the professional and yet fast review of our manuscript. We highly appreciate the time and effort you have invested in providing us detailed feedback, that helped improve the quality of the manuscript. In the following pages, we reply to the comments on a point-by-point basis.

We prepared responses to each of your concerns in the following in green, highlighted the changes in the manuscript in blue and look forward to the upcoming steps.

The authors,

*Julia Gebauer*
*Felix Prigge*
*Dominik Ahrens*
*Lars Wein*
*Claudio Balzani*

On the influence of cross-sectional deformations on the aerodynamic performance of wind turbine rotor blades (manuscript wes-2024-91)

**To all Referees:**

1. The introduction has been fundamentally revised. In this context, new references were added.
2. Section 2 "Methods" has been restructured.

**Referee 1:**

**General comments**

This manuscript presents an analysis of cross-sectional deformations in wind turbine blades under operational conditions and their impact on aerodynamic loads. The added value of this work is to quantify the aero-structural couplings where the structural cross-sectional deformations impact the airfoil shape, which in turn affects the aerodynamic loads. This field of study is relevant for the wind energy community due to the increase in blade flexibility. Furthermore, quantifying couplings is particularly important for multi-disciplinary design optimization of wind turbines.

The submitted manuscript presents weaknesses regarding (i) its contextualization with the literature, (ii) the strength of its methodology and (iii) its clarity and structure.

The literature review presents relevant and related references. However, it does not provide sufficient argumentation to motivate the work, i.e. justify why one could expect cross-sectional deformations to have a significant impact on aerodynamic loads. The review can be improved in several ways described in the specific comments below, in order to contextualize the study better and highlight its scientific relevance.

We agree that the literature review was weak. We significantly improved the introduction according to your specific comments. Please find more detailed explanations below.

The methodology used in the article consists of a sequential process. Aerodynamic loads calculated from an undeformed turbine are applied to a finite element model of the blade to obtain a deformed configuration. This deformed configuration is in turn used to update the aerodynamic characteristic of the airfoil and enables the calculation of the aerodynamic loads for the deformed turbine. This analysis is applied to a single load case, with constant wind and at rated wind speed. Furthermore, the analysis is done for one turbine, the 15MW reference wind turbine. Finally, the torsional deformation of the blade is not taken into account in the aero-elastic simulations. As pointed out by the authors in the manuscript, this methodology has weaknesses and may justify the lack of significant results. This weakens significantly the relevance of the study. Consequently, the work would be significantly improved by either (i) providing a thorough justification for the chosen methodology, (ii) applying the analysis to other wind turbine designs (e.g. 10MW and 22MW reference turbines), or (iii) improving the methodology to obtain positive results.

Yes, we agree that it would be important to extend the study by the proposed aspects. However, this is part of intensive work in progress and will be subject of (a) future publication(s). Please find more detailed explanations below.

Finally, several parts of the manuscript lack structure, clarity and conciseness. Several suggestions to improve this aspect are described in the specific comments below.

We improved structure, clarity and conciseness according to your specific comments. Please find more detailed explanations below.

On the influence of cross-sectional deformations on the aerodynamic performance of wind turbine rotor blades (manuscript wes-2024-91)

Specific comments

1. Literature review and contextualization.

- The literature review (l. 38-78) reads as a list of disconnected works, where the scientific contributions are not stated. Consider highlighting the problems that each cited work is addressing, and describing the associated results. Please highlight the similarities and differences between the cited work and the problem addressed in the manuscript. For the paragraph between l.38-56, linking the studies to the topic of blade cross-section deformation is critical to put the presented work in context.

- l.58: "Deformation of rotor blades ...": This paragraph is unclear. The first sentence mentions rotor blades and blade design, but it is not obvious how the cited works relate. Please make the link between the cited literature and the problem addressed by the manuscript more explicit.

As stated above, the introduction was thoroughly revised. A paragraph on experimental work as well as a paragraph on cross-sectional deformations due to icing, erosion, etc. has been added.

The respective text that we included reads:

In the meantime, blade tests revealed that cross-sectional deformations occur in rotor blades. Haselbach et al. (2016) have shown experimentally and numerically that critical bending moments lead to cross-sectional deformation and thus an opening of the trailing edge. Eder and Bitsche (2015b) conducted the fracture analysis for the trailing edge bonding of rotor blades and were able to show that in-plane deformations lead significantly to damage in the trailing edge adhesive. Jensen et al. (2012) studied the box girder experimentally and numerically. Here, the stresses in the shear webs could be measured as a result of the cross-sectional deformations.

A change in geometry of the aerodynamic shell and thus the airfoils can result from various aspects. Rosemeier and Saathoff (2020) investigated the change in geometry due to thermal residual stresses arising from the cool-down during manufacturing and the subsequent changes in lift and drag coefficients. They pointed out that the lift-to-drag ratios change particularly in the inboard regions of the blade. Simulations showed that the loads decreased and thus the fatigue life increased, which was accompanied by a reduction of power production. Leading edge erosion can also change the airfoil geometry due to removal of material. Gaudern (2014) investigated two cross-sections with different relative thicknesses in wind tunnel experiments. An increase in drag and a decrease in lift resulted in a significant decrease in overall performance. The airfoil geometry is also influenced by icing. Etemaddar et al. (2014) showed that the drag coefficient increases significantly, while the lift coefficient slightly decreases. Additionally, the authors compared thrust and power output for different wind speeds. They reported a shift of the power curve towards higher wind speeds, resulting in a higher rated wind speed and a lower power output in the partial load region. It can therefore be concluded that it is important to know possible cross-sectional changes and their impact on the performance of the wind turbine.

- l. 65-70: "Preliminary work on [...] and the deformed cross-sectional shape was obtained." This part of the literature review is critical for the motivation of the presented study. The authors cite two of their previous works that have quantified cross-sectional deformations in wind turbine blades. However, the interpretation of these works is weak. The authors only state the method used in these works, and not the results of the analysis. What is missing in the reasoning is a quantification of such deformations. In other words, were there enough deformations obtained in the cited works to suggest an impact on aerodynamic properties and justify the present work?

The preliminary work examined how the rotor blade deforms under different load scenarios. Overall, the deformations are relatively small. To which extent these geometrical changes have an impact on the

aerodynamic properties is not known and needs to be investigated. This is the subject of the manuscript. To make this clearer, we have included the following text in the introduction:

The deformations on the cross-section level were generally relatively small. However, it is not clear if small cross-sectional deformations can already influence the aerodynamic behaviour of the blades and with it have an aero-elastic coupling effect. This holds especially since, to the best knowledge of the authors, the load-induced impact of cross-sectional deformations on the aerodynamic behaviour of the airfoils and the coupling with the aero-elastic response of the blades was not yet investigated by other authors.

- Consider adding literature on the topic of multi-disciplinary analysis and optimization (MDAO), and aero-structural couplings (e.g. bend-twist coupling). The wind turbine design research community already includes such couplings in state-of-the-art design optimization frameworks. It would be relevant to highlight to which extent the state-of-the-art in MDAO takes into account cross-sectional deformations. See for example the following two references:

  o Mangano, M., He, S., Liao, Y., Caprace, D. G., & Martins, J. R. (2022). Towards passive aeroelastic tailoring of large wind turbines using high-fidelity multidisciplinary design optimization. In AIAA SCITECH 2022 Forum (p. 1289).

  o Bortolotti, P., Bottasso, C. L., Croce, A., & Sartori, L. (2019). Integration of multiple passive load mitigation technologies by automated design optimization—The case study of a medium-size onshore wind turbine. Wind Energy, 22(1), 65–79. https://doi.org/10.1002/we.2270

Thank you for referring these two papers.

Bend-twist coupling is an interesting technique to mitigate high aerodynamic loads in rotor blades and to reduce pitch activities in individual pitch control strategies for load reduction. However, it is out of scope of the manuscript, as it does not address the in-plane cross-sectional deformations investigated in the paper.

Cross-sectional deformations are addressed naturally when high-fidelity structural models are used in the context of load estimations via turbine simulations. However, when doing so, it is not possible to see the difference when these cross-sectional deformations are considered or neglected. Moreover, coupling high-fidelity models (3D FEM and CFD) for load simulations is far from being state of the art, as it requires enormous computing power and computation times. The simulation of whole design load cases does not seem feasible in the next years or decades.

- Consider adding literature on the topic of cross-sectional deformation in the context of failure analysis, including experimental works. This can help contextualize your results and quantify the deformations expected for operational loads vs ultimate loads.

Additional literature on this topic was included in the manuscript.

- Consider making it more explicit that the study focuses on cross-sectional deformations due to operational loads and not ultimate or failure loads.

The specification „operational loads" was added to improve clarity.

2. The description of the problem, research objectives and research questions is not clear in the text. As such, the text does not convey efficiently the relevance of the work.

- l. 36-37 "When optimising the design with the minimisation of the blade mass as an optimisation target, an increasingly elastic behaviour of the blade is expected, including the elasticity and

flexibility of the cross-sections": This statement introduces the problem addressed by the manuscript. Consider describing why an increase in cross-section elasticity and flexibility is a problem.

We added a sentence for clarification to the manuscript:

This also means, however, that the aerodynamically designed shell, which is reinforced by the structural components of the blade, is more susceptible to changes in the overall geometry during operation.

- l. 72: "Hence, the main objective of this paper is to study the influence of cross-sectional deformations in the rotor blade on the aero-elastic behaviour of the wind turbine.". Consider making the research question of the study explicit. For example: "To what extent do cross-sectional deformations impact the aerodynamic performance of the rotor, under operational conditions?"

To highlight the central research question, we rephrased this part in the manuscript:

Since normal operation is the most common condition in the life of a wind turbine, the following research question arises: To what extent do cross-sectional deformations affect the aerodynamic performance of the rotor under operational conditions?

3. Choice of methodology

- l.73 "A simple test case with respect to normal operation of the wind turbine and a constant wind field is selected for the initial investigation presented in this paper.": Please add a motivation for this choice of analysis.

In the lifetime of a wind turbine, normal operation conditions occur most frequently. Extreme load conditions can arise and show higher loads, but these cases are rare. By using a constant wind field, we do not introduce dynamics from the wind field, so that we observe a more quasi-static behaviour with respect to cross-sectional deformations, which allows a simplified 2-stage analysis methodology applied in the paper. In extreme load cases, we normally observe transients that introduce unsteady effects in the turbine, which requires a dynamic model for the cross-sectional deformations.

We included the following phrase motivating the focus on normal operation in the research question:

Since normal operation is the most common condition in the life of a wind turbine, the question arises: To what extent do cross-sectional deformations affect the aerodynamic performance of the rotor under operational conditions?

- l. 84 "A two-stage process is applied": please justify this choice of analysis. The state-of-the-art in MDAO does fully-coupled aero-structural analysis. Why has a sequential approach been chosen here and not a fully coupled one.

A fully coupled aero-structural analysis certainly has its advantages for the investigation of cross-sectional deformations. Calculating the cross-sectional deformations and the resulting lift and drag coefficients for each time step, however, means a high computational effort.  In Mandano et al. (2022) we read that a coupling of FEM and CFD was used for blade optimisation. Here, even a simple simulation with a steady state load case and no turbine dynamics requires a simulation time of up to 16 hours on a high performance cluster. In Caprace et al. (2022), (DOI: 10.2514/6.2022-1290) the extreme and fatigue load were first calculated with OpenFAST before using MDAO. It also states that it is not feasible to run the full design with these high fidelity models due to high computational cost.

**On the influence of cross-sectional deformations on the aerodynamic performance of wind turbine rotor blades (manuscript wes-2024-91)**

Our manuscript explicitly investigates the phenomenon of cross-sectional deformations. We aim for the future to derive a relation between loads and cross-sectional deformations to be able to take into account in low- to mid-fidelity turbine simulation tools and thus in blade design.

- l. 160-161 "torsion was not accounted for": this is a strong assumption for the study. Please justify why this modeling assumption was chosen, and its impact on the results of the study. There are several state-of-the-art aero-elastic tools that implement torsional degree of freedom. This is particularly critical in the light of the following statement: l. 405-406 "It is expected that especially torsion can result in significant in-plane cross-sectional deformations". If that is the case, why not include torsion in the first place in the study?

We assumed in the paper that the cross-sectional deformations are governed by the so-called Brazier effect. The Brazier effect relates cross-section ovalisations to bending moments. Hence, we considered the coupling between cross-sectional deformations and bending moments in the first place. However, there may be an additional effect from torsion moments. The evaluation of cross-sectional deformations from torsion is work in progress and will hopefully be published soon. We would like to emphasize that we do not refer torsion twisting to cross-sectional deformations, but in-plane warping within the cross-section plane. Hence, it is not just employing torsional degrees of freedom to calculate the cross-sectional deformation and the respective aeroelastic effect.

- l.163 "The loads were applied via multi point constraints that represented a load introduction similar to load frames". Please justify this model choice and its impact on the results of the study. Why simulate a blade-testing environment and not an operational environment, i.e. loads continuously distributed along the blade? This is particularly important considering the results of Figure 11. It seems that the high deviation observed at the location r/R = 0.8 is due to the presence of a load frame.

There are various approaches for load application in FE rotor blade models. Can et al. (2020) apply the extreme loads at the nodes of the spar cap of the pressure and suctions side along the blade. Haselbach et al. (2015) apply the loads to a master node in the middle of the spar caps with coupling constraints to the other nodes of the spar caps. Our approach is to use loads that are concentrated in the longitudinal direction but distributed across the cross-section. All approaches have in common that they approximate the continuous bending moment distribution along the blade. Our approach was validated in comparison with a full-scale blade test, see Noever-Castelos et al. (2021), where the load introduction is comparable to our load frame-like modelling approach. However, the bending moments are approximated quite well, and thus it is assumed that the Brazier effect is approximated good enough. Since in the postprocessing evaluation we only consider regions that are unaffected by the load introduction, the methodology seems reasonable. We would like to emphasise that every modelling approach comes with the risk of boundary condition effects. In our case we at least ensure a reasonable distance to load introduction points to not provoke local effects.

For carried out additional analyses considering the load frame no. 5 (LF5), where we moved the position of LF5 towards the blade tip to investigate the cross-sectional deformation close to that load frame. The following figure shows the relative deviation of the chord length over the blade span.

[Figure]

The blade root (here LF1) is fully clamped. Load frames 2 – 4 are fixed for all simulations. We changed load frame 5 over 5 different positions. It can be seen that the peak between 90–95 m blade span occurs for all 5 load frame positions. The magnitude of the peak changes, however.

- l.188 "High-fidelity computational fluid dynamics (CFD) was used to compare the polars and ensure the validity of XFOIL results." Why use XFOIL for the generation of airfoil polars and not CFD directly?

CFD analysis for all cross-sections would result in extensive computational effort, which does not seem to be justified in view of the very good agreement between CFD and XFOIL results for the two analysed cross-sections. For the analysis of 3D effects associated with the flow, the use of CFD is considered for the future.

4. Concise description of the method. Section "2. Methods" starting at l.80 presents the methodology of the work. The different steps of the analysis are reported: (i) Generation of loads using aero-structural simulations for the undeformed turbine, (ii) Generation of cross-sectional deformations from the extracted loads using a FE model of the blade and (iii) Generation of loads for the deformed configuration. However, the text lacks structure and conciseness.

- The text describes in length adjacent aspects of the analysis that are not central to the study: creation of a FE blade model from the reference turbine data (l.85-91, l.109-111), verification of the FE blade model (l.112-135), verification of the airfoil polars between XFOIL and CFD (l. 190-215), choice of the number and placement of load frames (l.170-173). Consider moving these parts of the text into appendices.

The sections on the verification of the FE model and the polar calculation were shifted to the appendices A and B, as suggested by the referee. However, the creation of the FE model is an important step, as the cross-sectional deformations, which is the core of the paper, are calculated with it. Moreover, the explanations on the number and positioning of the load frames is important to understand how we have ensured the existence of valid deformations along the entire blade and to provide reproducibility of the research results. We have thus kept the respective sections in the main text.

- Consider restructuring section 2. Due to the fact that several models are used twice in the analysis, I would recommend first describing the steps of analysis with the support of Fig.1 and then including one section for each model used. For example: "2.1. FE structural model", "2.2. Generation of airfoil polars", "2.3. Aero-elastic simulations". Then, each section can describe when the associated model is used in the global process. For example: "Aero-elastic simulations are conducted during steps 1 and 2 as shown in Fig. 1, using the undeformed and deformed configuration of the turbine, respectively."

The structure proposed by the referee also was our original idea. Obviously, we did not succeed in a clear realisation. We have restructured section 2 according to the referee's recommendation, with little adjustment in the subsections' titles.

- Having a flowchart like Fig. 1 to illustrate the different steps of the analysis is great to support the description of the methodology. However, there is a discrepancy between the terms used in the figure and the terms used in the text, which can be confusing for the reader. To increase the relevance of the figure, consider aligning such terms and referring to the headers of section 2 in the figure.

&

- Some steps of the analysis are repeated excessively in the manuscript. For example, the use of OpenFast for aero-elastic simulations is mentioned in l.94-95 ("an aero-servo-elastic simulation of the turbine was performed using OpenFAST"), l.157 ("an aero-servo-elastic turbine simulation with OpenFAST"), l.228 ("The aero-servo-elastic simulations were performed with OpenFAST"), l.256 ("Aero-servo-elastic simulations with a duration of 600 s were carried out in OpenFAST"). Consider making the description of the analysis more concise and referring to section 2. when needed.

For a more concise description, we stick to a more general formulation in section 2. In the process depicted in Fig. 1, the above-mentioned repetitions have been deleted and the same terms were used.

- l.247-254 "Recall the blade element theory... have an impact on the lift and drag forces.": The equations and notations introduced in this paragraph are not used in the rest of the manuscript. In order to improve the conciseness of the text, consider removing this part or moving it to a part of the manuscript used to motivate the work.

The equation is introduced to show which variables are relevant for the load calculation and to motivate the choice of features looked at in the following subsections. It is thus important to let the reader follow the order of thoughts. However, it is not necessarily important for the overall motivation of the research, but for specific parts of it. Hence, we prefer to leave the equation where it is and kindly ask for your agreement.

4. The results of the study are described in sections "3. Cross-sectional deformations" and "4. Coupling effect on the turbine behaviour". The documentation of the results is thorough but lacks interpretation and conciseness.

- l.246 "the cross-sectional deformations at two positions along the blade are presented and discussed" and l. 319 "The cross-sectional deformations were determined for all cross-sections along the blade. However, due to space limitations, they cannot be discussed in detail here.". The distribution of the cross-sectional deformation would be relevant for the scientific community and for reproducibility of the results. Consider shortening section 3.1. and adding one figure and one paragraph describing the deformations across the blade.

A single figure that shows and highlights the cross-sectional deformations for a whole rotor blade is difficult, as it changes not only quantitatively, but also qualitatively. For improving the reproducibility of results, we will consider providing additional datasets on a repository including data on the deformed cross-sections along the blade once the paper is accepted for publication and before final publication of the manuscript. The datasets will then be referred to in the „Code and data availability" section at the end of the manuscript to make them findable. We will clarify the required process with the editor.

- Consider combining Figures 7 to 10 to help the reader see the differences between each case.

We combined the figures 7-10 to one figure, as suggested by the referee.

- In section 3.1, the authors describe at length Figures 7 to 10. However, an interpretation of the results is missing. For example, l.292-294 "From the leading edge up to approximately c/8, the shell on the pressure side deforms into the cross-section. From there on up to approximately 5c/8, the pressure side shell deforms out of the cross-section.": This is a visual description of the results. Please provide some interpretation: can the deformation of the cross-section be justified by the type of loads or the stiffness of the components? Are these results similar to deformations described in the literature? Furthermore, consider making the description of these results more concise and to the point. This comment also applies to sections 3.2 and 3.3

We discussed the results and added the following points to the manuscript:

Overall, it can be seen when comparing Fig. 8 (b) and 8 (d) that the cross-sectional deformation leads less to a loss in lift-coefficient and more to an increase in the drag coefficient. This is consistent with the literature (recall Sec. 1). The relative deviation in the drag coefficient allows the assumption that the resulting loads are greatest for β = 180°

- l.358-359 "Two regions along the blade span can be identified that show a change in aerodynamic properties. The first region is at around 20 % of the span, the second at around 80 % of the blade span.": This is an expected result considering the results shown in Figure 11. Please make the causality between the results of section 3.2 and section 3.3 clearer, i.e. the deformation shown in Figure 11 results in the change of aerodynamic characteristics in Figure 12.

For the calculations of the aerodynamic polars, the chord length of the airfoils was always normalized. However, we agree, that a one can see the change in chord length as a parameter for the cross-sectional deformation. Thus, a change in lift and drag coefficients is more likely to occur where the change in chord length can be noticed.

We added to the manuscript a short decription:

Based on the qualitative shape of the relative deviation of the chord length, we can derive where the cross-sections deformed the most. As the outer shell is the main factor in the calculation of the lift and drag coefficient, it can expected that the qualitative results will be similar.

- Figures 13 and 14 are particularly clear and efficient in representing the data.

Thank you, we are delighted to hear that.

- Consider linking the results of section 4 to the results of section 3 in a more explicit manner. Do the differences in lift and drag reported in section 3.2 justify the results of section 4?

We added a discussion to the in-plane and out-of-plane bending paragraph in Sec. 4:

Due to the relative deviation in chord length, it was assumed that the loads would show the greatest deviation at β = 90°. However, it turns out that the greatest change occur at β = 0° (+) and at β = 180° (-). This also seems plausible, as the relative change in the drag coefficient is of higher magnitude than the change in chord length.

Here, the maximum value shows for β = 90∘. We can see here a similar trend between out-of-plane bending moment and chord length deviation. A correlation to lift and drag coefficient, however, cannot be derived.

5. Discussion of the results

- l. 394 "This study revealed small changes ...": It could be relevant to have a separate section for the discussion of the results

In an older version of the manuscript, we separated the two sections as mentioned. In order to discuss the results head on, we decided later to merge these two sections.

- Consider contextualizing the results of section 4. Would the cross-sectional deformation have a significant impact on power production or other relevant metrics? What would be the impact for wind turbine design?

We agree. Considering the loads is of course interesting for the blade design, but a comment on power production, life time or perhaps new design approaches is the goal, however, this is not yet achievable. We hope to be able to address this in an upcoming publication.

6. Conclusion of the work

- l.14: "the initial results imply that further investigations should be carried out with more complex wind fields and different rotor blade designs" This statement needs to be backed up by either data or literature. Placing it in the abstract implies that the article supports this claim, which is not the case in this version of the manuscript. Please soften the language here, for example: "While these values are small, further investigations with ... could identify a stronger aero-structural coupling".

The manuscript was revised accordingly.

- l. 426-427 "The change in chord length was identified in the vicinity of 20 % and 80 % of the blade span.": Please put these results in context with the type of load applied, i.e. with discrete load frames and not a continuous loading.

Blade span of 20% describes the position of the cross-section with the maximal chord. Literature also shows that this is an area where cross-sectional deformation can be expected due to long flat areas of e.g. trailing edge panels.

At 80% blade span there is a chance that we hit a transition to buckling. This might occur due to applying the load frames in this area. However, the structural lay out.

We added the following paragraph to the manuscript:

This is also the range of blade span the load frames were applied. The cross-sections with maximun chord length are in the area. Here, the relative deviation of chord length indicates a change in cross-sectional shape, which is in accordance with literature. The maximal deviation of chord length at 80 % might be a result from buckling. A more detailed analysis would be required for a precise statement.

- l. 432-439: "These relative deviations are quite small. However, ... We therefore expect the influence on the aero-elastic simulation to be greater when analysing extreme load cases.": This paragraph significantly weakens the results of the study. In essence, the author states that the relative deviations reported in the study are small because simple assumptions were taken, which negates the relevance of the work. This is done to justify the "negative results" obtained. However, it raises the question of why such assumptions were used. An alternative conclusion to the work would be to suggest that the aero-structural coupling between cross-sectional deformation and aerodynamic

loads is negligible for operational loads. This would significantly increase the impact of the work and its relevance for the scientific community.

Thank you for pointing this out. It is a bit tricky to derive a general statement from one study. We have therefore reworded the conclusion. We want to clarify that the small changes apply specifically to the operational load case, which assumptions have been made and where we still see potential.

These relative deviations are quite small. It needs to be highlighted that this only holds for an operational load case. This is based on the assumption that the cross-sectional deformations are greatest when the flapwise and edgewise bending moments are at their highest, i.e. rated wind speed. This leads to the conclusion that the aero-structural coupling between cross-sectional deformation and aerodynamic loads is negligible for operational loads.

However, it must be taken into consideration that a number of simplifications have been made. E.g., the torsional moment was not included in the calculation of cross-sectional deformations. For a more precise statement regarding cross-sectional deformations, torsion is planned to be included in the near future. Furthermore, the wind field was simplified using a constant wind field at the rated wind velocity. Although the cross-sectional deformations were small in this case, it cannot be concluded at this stage that this holds for all possible operation conditions in normal power production. Hence, additional wind velocities need to be investigated, as a blade rotated by the pitch angle in over-rated conditions may experience different cross-sectional deformations, both qualitatively and quantitatively. Apart from that, extreme loads will likely result in more severe cross-sectional deformations, so that we expect the influence on the aero-elastic response to be greater when analysing extreme load cases.

7. Description of the methodology

- l. 6 "a 3D finite element (FE) model": please add details of the model in the abstract, for example stating the type of elements used.

The element type „shell elements" was added. The more detailed description follows in section 2.1

- l. 93 "... and were used to calculate the aerodynamic loads via the blade element momentum theory (Jonkman et al., 2015). For the load calculation, an aero-servo-elastic simulation of the turbine was performed using OpenFAST (National Renewable Energy Laboratory, 2023).": Please clarify whether the "aerodynamic loads" and "loads" mentioned in the two sentences are calculated with different tools, and if loads due to gravity are included in the analysis.

The cross-sectional deformations have an impact on the aerodynamic loads. These loads effect the beam model in OpenFAST. We extract the internal loads, i.e. bending moments from OpenFAST and apply them to the 3D FE blade model.

The manuscript was revised accordingly. The related text now reads:

For the load calculation (considering aerodynamic and gravitational loading), an aero-servo-elastic simulation of the turbine was performed.

**Minor comments**

- For the cited work "Gebauer, J. and Balzani, C.: Cross-Sectional Deformation of Wind Turbine Rotor Blades, in: The 33rd International Ocean and Polar Engineering Conference, Ottawa, Canada, 2023.", the full paper (if there is any) couldn't be found from the information provided. Only a one-pager was found, without mention of relevant results

(https://publications.isope.org/proceedings/ISOPE/ISOPE%202023/data/pdfs/160-2023-TPC-0416.pdf). Please provide the reference to the full paper or remove the reference.

We agree that it is not easy to find the paper instead of the abstract. We have included the full bibliographic details including the paper number . Note: The reference is not available open access.

- Consider adding an illustration of the cross-section topology in the manuscript.

In the process of revising section 2 "Methods", an illustration of the rotor blade and a cross-section was added to the manuscript, see Fig. 2.

- The term "MoCA model" is used repeatedly in the manuscript. The qualifier MoCA refers to a specific modeling tool, and would not be meaningful for readers not familiar with it. Consider using another term, such as "the 3D blade geometry model".

&

- The term "OpenFAST simulations" is used repeatedly in the manuscript. Consider using the term "aero-elastic simulations" instead to highlight the type of analysis conducted, instead of the tool used.

We agree that a more general formulation makes more sense here. During the process of restructuring chapter 2 "Methods", all specific descriptions were replaced by such.

- The authors use the past tense repeatedly in the manuscript. Please make sure to use past tense when describing your numerical experiments, and use present tense when describing your results. For example, l.326 "The relative deviation was plotted against the normalised spanwise coordinate": instead of using "was plotted", consider using "is shown" instead.

The manuscript was revised accordingly.

- l. 11. "depend largely": the qualifier "largely" does not add extra meaning here. Consider rephrasing to keep the abstract concise and to the point.

The manuscript was revised accordingly.

- l. 26 "which is calibrated by the aerodynamic twist angle". Please consider replacing the term "is calibrated by" with "depends on" in order to increase the precision of the text.

The manuscript was revised accordingly.

- l. 31 "In the context of this paper, the structural composition of the blade defines its resistance against cross-sectional deformations": please clarify or rephrase. This sentence is unclear.

The term „structural composition" was replaced by „structural design" to improve clearness.

- l. 35 "structural loads increase." Consider aligning the term used here with the load types enumerated in l. 19-20. This will help the reader understand exactly what type of load is meant by "structural loads".

The structural loads are understood as internal forces and moments in the structural members. We have clarified the formulation by the following phrase:

[...] and thus the structural loads, which are understood as the internal forces and moments in the blades and other structural members, increase.

- l. 41 "The study further focused on bifurcation analysis, which is not the subject of this paper.": Consider removing this sentence.

  The sentence was deleted in the revised manuscript.

- l. 59 "energy-based method for thin-walled ...": for modelling? analysing? designing? Consider adding a verb in this sentence to characterize the method.

  The sentence was deleted in the revised manuscript.

- l. 66 "a three-dimensional finite element model": please use "3D" or "three-dimensional" consistently throughout the manuscript.

  The manuscript was revised accordingly. The abbreviation "3D" is now introduced when the term "three-dimensional" is first used in the main text (except the abstract, where no abbreviations are normally used), and then "3D" is used consistently throughout the manuscript.

- l. 236 "The simulation was carried out until the rotor blade behaviour became periodic." and l. 256 "Aero-servo-elastic simulations with a duration of 600 s were carried out". These two sentences are not coherent with each other. Does the first sentence mean that the simulation was carried out until the end of the transient period? In addition, the 600s duration of aero-elastic simulation usually refers to a period of time after the transient period. Please precise the duration of the simulation, and whether it includes or not the transient period.

  The duration of the aero-servo-elastic simulation was 700 s. From this time span the first 100 s were removed as transient period. The remaining 600 s were used for the analysis.

  A more detailed description was added to the manuscript in Sec. 2.3.

- l. 329 "The first one is below 30%": consider changing the unit in Figure 11 to a percentage to match the description of the results in the text.

  The on-the-fly transfer of percentage numbers to relative numbers and vice versa is daily practise in engineering. Using one or the other is a matter of taste, but does neither improve nor deteriorate the content. Hence, we prefer to keep the figure as it is and kindly ask for your agreement.

- l. 403 "They are small, but they are finite." The term "finite" is generally used in opposition to "infinite". Consider using "non-negligible" instead.

  The manuscript was revised accordingly. The related text now reads:

  They are small, but non-zero and thus not negligible.

**Referee 2:**

The authors investigated the effect of blade cross-sectional deformations on the aerodynamic behaviour of a large-scale wind turbine, i.e., the IEA 15 MW RWT. The reviewer believes that the topic and the activity are very interesting, innovative and worthy of investigation. The study is overall of good quality and both methodology and results are presented in a rigorous and exhaustive way. Nonetheless, two main flaws are preventing it from being published directly and need to be revised:

A. In the Introduction, the authors struggle in highlighting the scientific question and the novelty of their study. Lines 39-46 in the manuscript, in particular, are not functional to the narration and only make the introduction more vague;

The introduction has been thoroughly revised. The beam theory paragraph was removed and new paragraphs on experimental work on cross-sectional deformations and the impact of thermal residual stresses, icing, etc. as sources of cross-sectional changes on the aerodynamic performance were added.

B. While the adopted FE methodology is rigorous and adequate for the scope of the study, the CFD one is inefficient and presents many limitations. In the Reviewer experience, a 3D approach is not necessary when using a RANS formulation. Switching to 2D would save you a lot of computational effort and allow to use CFD to produce all the polars necessary for the analysis. In this perspective:

This may be true for the prediction of static polars. However, we prepared the CFD model within a research project that focusses on dynamic stall prediction. According to Braud et al.,2024 (Study of the wall pressure variations on the stall inception of a thick cambered profile at high Reynolds number), the stall in a thick cambered profile is inherently local and three-dimensional, that is why the authors have chosen Q3D dimension instead of 2D. As mentioned before, we prepared the domain not only for the static performance, but also the dynamic stall studies, where the spanwise extension of the domain is necessary. New studies of our own confirm the need for Q3D setups (Kim et al., 2024, DOI: 10.5194/wes-2024-31), for this profile and Reynolds-Number. As investigating the impact of cross-sectional deformations on the dynamic stall behaviour of the airfoils is one of the next steps, using the same model as will be required for that purpose seems reasonable. Moreover, given the very good agreement between XFOIL and CFD results, it does not seem necessary to use CFD for all cross-sections.

1. the adopted domain dimensions are too small for the Reynolds numbers under consideration

To the best of our knowledge 50 times the chord length is sufficient. Vitulano et al. (https://doi.org/10.5194/wes-2024-47) tested a ratio of 10, 30, and 50 for the same profile at a Reynolds Number of 1.8 10^6, showing no relevant difference of the domain size. Therefore, they proceeded with a ratio 10 only. So, we assume that using a ratio of 50 is a conservative simplification. If the reviewer has further insight in this context, we would appreciate to receive a reference and we will investigate the influence next.

2. a proper mesh sensitivity analysis should be performed

A mesh sensitivity analysis is a natural thing in CFD analyses. It has been done in Ahrens et al. (2022) for a similar case (but smaller Reynolds-Number) and in Kim et al., 2024 (DOI: 10.5194/wes-2024-31) for the same case. Both studies indicate that our mesh resolution is sufficient especially for small angles of attack (below 10.) Vitulano et al. (https://doi.org/10.5194/wes-2024-47) conducted a mesh dependency study, showing almost no sensitivity between 74 10^4 and 12 10^4 cells (for the same profile at a Reynolds Number of 1.8 10^6) in the 2D domain. In Kim et al., 2024 (DOI: 10.5194/wes-2024-31) we investigated meshes with a similar resolution. An exact comparison with Vitulano et al. is not possible because the latter does not report non-dimensional cell sizes and they used a wall-function approach. To conclude, similar studies showed that the mesh chosen in this paper is sufficient, especially for low angles of attack. If required, we will extend our existing mesh dependency study and include it into this paper. We would like to emphasize that the CFD part in this paper is for verification of XFOIL results only. The further analysis is based on XFOIL results. Hence, the description of the CFD solution was shifted into the appendix according to the first referee's suggestion. It does not seem reasonable to extend the CFD appendix by a mesh sensitivity analysis.

3. not considering laminar–to–turbulent transition effects might be detrimental for the prediction of both absolute and relative loads

Laminar–to–turbulent transition is considered in the XFOIL calculations. Since the boundary layer transitions to turbulent very near the leading edge, fully turbulent assumption in CFD seems valid from our point of view. This was experimentally shown by Kiefer et al. 2022 (Dynamic stall at high Reynolds numbers induced by ramp–type pitching motions) and further justified by Kim et al. 2024 (DOI: 10.5194/wes-2024–31).

4. your CFD model should be validated against experiments. I suggest the database from the AVATAR project

We agree but when we started the project, we had no experimental data available. So, thanks for pointing us to the AVATAR project. Quick study showed, that it covers dynamic stall at least up to a Reynolds–Number of 2.5 million, for mostly thin and symmetric profiles. We are focussed on Reynolds Numbers above 3 million with thick and non–symmetric profiles, which changes the aerodynamic characteristics regarding laminar to turbulent transition and dynamic stall. Anyway, the database will be studied in more detail and will be considered for the future. For the near future we have planned a validation based on the experiments from Kiefer et al. 2022.

Minor revisions:

1. Section 2 has a lot of information that is repeated in the following paragraphs. Please revise it to avoid repetition;

Section 2 "Methods" has been revised accordingly. A more general description was given that fits to the flow diagram. The individual steps are the given in more detail in subsections. Full details of the verification have been moved to the appendix.

2. Figures 3-5: please indicate in the caption what the different frames are, using letters or "left" and "right"

Letters (a,b,c,...) have been inserted into all multi-paneled figures for a clear identification.

3. It would be nice to add a figure representing the FE model

We added figure 2 to the revised manuscript. Here, you can see the blade FE model. However, due to visualisation matters, the mesh is coarser than in the simulations.

4. Figure 4: the right picture is redundant, I suggest adding some details of the mesh instead

Thank you for the suggestion. We included in Fig. B1 two zoom views of the mesh.

---

## Author Response (AR2)

**RESPONSE TO REVIEW**

Thank you for your support in improving this manuscript.

We prepared responses to each of your comments in the following in green, highlighted the changes in the manuscript in blue and look forward to the upcoming steps.

Additionally, the manuscript was checked and edited for grammar and spelling mistakes.

The authors,

*Julia Gebauer*
*Felix Prigge*
*Dominik Ahrens*
*Lars Wein*
*Claudio Balzani*

**Report #1**

Thank you for the time and effort you put into the revision of the manuscript. As a last modification, I recommend adding some details about the mesh – in the same fashion of your response to my review – to the new Appendix B. Regarding the domain dimensions, please have a look at this work from Sørensen et al: https://iopscience.iop.org/article/10.1088/1742-6596/753/8/082019

As recommended, we added the discussion about and a figure of the mesh to the paper (Appendix B):

l. 464 – 474: Vitulano et al. (2024) conducted a mesh dependency study for the FFA-W3-211 profile at a Reynolds Number of $1.8 \cdot 10^6$, showing almost no sensitivity between $12 \cdot 10^4$ and $74 \cdot 10^4$ cells in the 2D plane. Kim et al. (2024) presented a mesh sensitivity study for the FFA-W3-211 profile at a Reynolds-Number of $15 \cdot 10^6$, showing that further refinement of the mesh in spanwise direction has a negligible influence on the prediction of the dynamic stall cycles. They used $35 \cdot 10^4$ cells in the 2D plane with 20 cells in spanwise direction of the 3 dimensional model. A more precise comparison with Vitulano et al. (2024) is not possible because they don't report non-dimensional cell sizes and they used a wall-function approach. A mesh dependency study for slim profiles at a Reynolds-Numbers of $0.4 \cdot 10^6$ was done by Ahrens et al. (2022), showing that a mesh with $y+ \leq 1$ and $x+ \leq 700$ is sufficient to investigate integral blade loads during a dynamic stall cycle. So, the cell dimensions used in this paper are assumed to yield mesh independent predictions of lift and drag coefficients, especially in the linear region of XFOIL.

We appreciate the link to the paper from Sørensen et al. We will test the influence of even larger domain sizes, i.e. more than 50 time the chord length in the future. In our past sensitivity studies we did not see a relevant change of results above 50 times the chord length and accepted the "uncertainty" as a compromise between accuracy and efficiency.

**Report #2**

The authors provided a thorough and well argued reply to the reviews. The contextualization and scope of the study has been strengthened significantly. However, the interpretation of the results can still be improved, as well as the clarity and conciseness of the text.

We have further revised our manuscript in this regard. Please see below for more details.

In this context, the reviewer recommends the following minor revisions:
– The main message of the study needs to be more clear and precise. The manuscript highlights clearly the weaknesses of the study, and state that future work is needed (see l.405-415). In this context, the relevance of the present work is significantly undermined. In other words, why one should read this paper, instead of waiting for the next one? What is the added value of the present work? This comment can be addressed by thorough interpretation of the results in the light of the methodology used.

We have worked on clearly formulating our main message: The influence of cross-sectional deformations in wind turbine rotor blades on internal loads are marginal for operational conditions with maximum loads in flap- and edgewise direction (i.e., operational loads at the rated wind speed).

This statement was used for the abstract and conclusion. We also went through the manuscript in order to clearly emphasise the main message.

l. 13 – 14: These results show that cross-sectional deformations have a minor influence on the internal loads of rotor blades in normal operation.

l. 395 – 406: The aforementioned relative deviations are associated with a load scenario of normal operation at the rated wind speed and thus with maximum operational loading (i. e., maximum

thrust and torque). Nevertheless, the deviations in root bending moments are quite small. It can thus be concluded that the impact of cross-sectional deformations on the aero-elastic response of the turbine is small (and potentially negligible) for normal operation. However, it should be noted that this conclusion is design-specific, meaning that the aero-elastic effect can be more pronounced in other turbines depending on the design philosophy with respect to the blade's stiffness. To derive more general conclusions, different turbines and rotor blades could be analysed. Moreover, higher degrees of loading, e. g., from extreme load cases, may result in higher cross-sectional deformations and thus to higher aero-structural couplings. Hence, a broader variety of load scenarios including combined loading with torsion from extreme conditions should be investigated in future work. Independent of particular results, the methodology presented in this paper could be used to numerically verify the absence of potentially undesirable aero-elastic couplings originating from cross-sectional deformations during the design, which could help to increase the reliability of wind turbines in the future.

– The text needs to be carefully edited to improve the conciseness of the manuscript. This is particularly important for the presentation of the results, which is at times lengthy and convoluted, and prevents the reader to capture the essence of the work efficiently.

&

– There is still a lack of connection between the results presented. The authors have only partially addressed my comments on this aspect. To be more precise, having more connection between results means that after reading sections 3.1 and 3.2, the reader can expect the following results. In addition, it means that the results of section 3.4 can be fully understood from the previous sections. This will help with the clarity of the manuscript.

To make the manuscript more concise, we moved some paragraphs in the results section and shortened them. The descriptions in particular are shortened and more attention is paid to the explanations and connections. The modifications can be seen in the manuscript with highlighted modifications (see LatexDiff document). We hope that we have now addressed your comment appropriately.

**Minor comments:**
– The references should follow WES guidelines. For example, in-text references are cited without parentheses (see l.75)

Thank you for pointing this out. We checked the entire manuscript carefully on citation style.

– l.3-4: "an initial estimation of the extent to which cross-sectional deformations influence the aerodynamic load distribution along the rotor blade." The presented result show results for blade root bending momentum, and not load distribution. Consider replacing the end of the sentence to be more precise, e.g. "... influence the aerodynamic loads on the rotor".

Manuscript was revised accordingly.

– l. 14-16: "Although these values are relatively small, the initial results imply that further investigations should be carried out with more complex wind fields and different rotor blade designs to identify aero-structural couplings that could potentially be critical for the design of rotor blades or other wind turbine components." Please align this statement with the conclusion of the manuscript.

This part was revised. It states now:

l. 13 – 14: These results show that cross-sectional deformations have a minor influence on the internal loads of rotor blades in normal operation.

– l50-51: This paragraph does not seem to be connected to the adjacent paragraphs. Consider adding some glue text or merging this part of the literature review with another paragraph.

This section has been merged with the following paragraph.

– Eq. 1: This equation presents the equation for the relative difference. However, the symbol is not used in the rest of the manuscript. Furthermore, the equation for relative difference is common knowledge. Please remove this equation for conciseness.

We removed the equation and the text parts in which the equation was referenced.

– Figure 4: "The maximum sampling error is 1.84e−2": Please use the notation 10^(-2) and not e^(-2).

Manuscript was revised accordingly.

– Figure 9 and 10: The cut-out on the main figure do not match the cut-outs on the subfigures, which is confusing. For example, for Fig. 9 (a), the cut-out on the main figure seem to span [-20 degrees, +20 degrees] and [0.4 kNm, 0.9 kNm], whereas the cut-out figure has a much larger span. If the cut-outs are not a zoomed version of the main figure, it reduces their significance significantly.

Figure 9 and 10 were revised accordingly. We deleted the zoom figures, as they did not present results other than those shown in the non-zoom figure. We have included the relative errors in the non-zoom figure instead.